# Evolutionarily unique mechanistic framework of clathrin-mediated endocytosis in plants

Madhumitha Narasimhan[1†], Alexander Johnson[1†], Roshan Prizak[1,2], Walter Anton Kaufmann[1], Shutang Tan[1], Barbara Casillas-Pérez[1], Jiří Friml[1]*

[1]Institute of Science and Technology Austria, Klosterneuburg, Austria; [2]Institute of Biological and Chemical Systems - Biological Information Processing, Karlsruhe Institute of Technology, Eggenstein-Leopoldshafen, Germany

**Abstract** In plants, clathrin mediated endocytosis (CME) represents the major route for cargo internalisation from the cell surface. It has been assumed to operate in an evolutionary conserved manner as in yeast and animals. Here we report characterisation of ultrastructure, dynamics and mechanisms of plant CME as allowed by our advancement in electron microscopy and quantitative live imaging techniques. *Arabidopsis* CME appears to follow the constant curvature model and the *bona fide* CME population generates vesicles of a predominantly hexagonal-basket type; larger and with faster kinetics than in other models. Contrary to the existing paradigm, actin is dispensable for CME events at the plasma membrane but plays a unique role in collecting endocytic vesicles, sorting of internalised cargos and directional endosome movement that itself actively promote CME events. Internalized vesicles display a strongly delayed and sequential uncoating. These unique features highlight the independent evolution of the plant CME mechanism during the autonomous rise of multicellularity in eukaryotes.

*For correspondence:
jiri.friml@ist.ac.at

†These authors contributed equally to this work

Competing interests: The authors declare that no competing interests exist.

## Introduction

Clathrin mediated endocytosis (CME) is the key cellular progress of internalising a cargo, either on the plasma membrane (PM) or outside the cell, into the cell. The cargos are packaged inside a small vesicle, which is covered by a coat made of the protein clathrin. CME represents a major endocytic route in eukaryotes (*Goh et al., 2010*). In plants, it is even more significant as most, if not all, endocytosis occurs via CME, and many important PM proteins implicated in key physiological processes, such as growth and development, nutrient uptake or pathogen defence, are established cargos for the CME pathway (*Barberon et al., 2011*; *Dhonukshe et al., 2007*; *Di Rubbo et al., 2013*; *Mbengue et al., 2016*; *Yoshinari et al., 2016*). Therefore, CME in plants represents a major mechanism for regulating both local cellular signaling and the global response to external stimuli.

CME in plants is poorly characterized and its hypothesized mechanism is largely inferred from studies in mammalian and yeast systems, where CME components are highly conserved; with at least 60 homologous endocytosis accessory proteins (EAPs) well characterized in both these model systems (*Kaksonen and Roux, 2018*; *Lu et al., 2016*; *Merrifield and Kaksonen, 2014*; *Taylor et al., 2011*). Plants possess homologues of the many of the core EAPs, for example, there are two clathrin heavy chain proteins (CHC 1 and 2) and three clathrin light chains (CLC 1, 2 and 3, which have at least 30% sequence homology with mammalian CLCa and CLCb *Wang et al., 2013*), adaptor proteins (e.g. all the AP2 subunits, AP180), dynamin like proteins (e.g. Drp1s and Drp2s) and uncoating factors such as Auxilin-like proteins (*Adamowski et al., 2018*; *Baisa et al., 2013*; *Chen et al., 2011*). However, functional characterisation of these, and other plant EAPs, is greatly lacking.

Despite a high degree of conservation between mammalian and yeast EAPs, there are key differences in the actual CME mechanism; chiefly related to the requirement of actin cytoskeleton. While actin co-localises with single sites of CME in both systems, interference with actin results in a total arrest of CME in yeast (*Kaksonen et al., 2005*) whereas in mammalian cells, there is only an overall reduction of endocytosis but still ongoing productive CME events (*Fujimoto et al., 2000*; *Merrifield et al., 2005*; *Taylor et al., 2012*; *Yarar et al., 2005*). This difference has been attributed to the higher turgor pressure of the cell wall-encapsulated yeast cells (*Aghamohammadzadeh and Ayscough, 2009*; *Boulant et al., 2011*; *Dmitrieff and Nédélec, 2015*). As plant cells are subject to equal or even higher turgor pressures compared to yeast cells (*Beauzamy et al., 2014*; *Schaber et al., 2010*), actin has been hypothesized to be crucial for plant CME; nonetheless, it has not been assessed in great detail.

Thus, despite the crucial physiological and developmental importance of CME in plants, little is known about its actual mechanism and how it differs from the well-characterized CME models in mammalian and yeast systems. One of the major reasons for this is the inability to directly observe CME and clathrin coated vesicles (CCVs) *in vivo*. For example, electron microscopy, the key technology credited for the first CME characterisation (*Roth and Porter, 1964*; *Schmid et al., 2014*), can be performed with plant tissues (*Robinson, 1996*; *Valk and Fowke, 2011*) but the successful capturing of ongoing CME events is rare. It is only recently that the plant field has started to adopt live imaging methods to directly look at the dynamics of CME events; like total internal reflective microscopy (TIRF-M) (*Johnson and Vert, 2017*; *Vizcay-Barrena et al., 2011*).

Here we provide a detailed characterisation of plant CME, starting with the formation of clathrin-coated pits (CCP) at the PM and up until the later endocytic events in the cell interior. We established new electron microscopy sample preparation protocols, which produced an abundance of clathrin-coated structures at an unprecedented level *in vivo*, and combined this with live quantitative TIRF-M imaging at a high spatial and temporal resolution. Contrary to expectations, actin is not present or required during CCV formation on the PM, but is critical for the early post-endocytic traffic supporting an unsuspected active role of dynamic actin and moving endosomes in organizing CME events at the PM. Imaging of CCVs deeper inside the cell revealed strongly delayed and gradual uncoating on route to the endosomes. Thus, we provide an extensive mechanistic framework for CME in plants and identify multiple unexpected, evolutionary non-conserved aspects of this fundamental cellular process.

## Results

### Different populations of clathrin-coated structures at the plasma membrane and intracellular membranes

Endocytic events have not yet been well characterized *in planta*, largely due to the lack of efficient ultra-structure analysis of the key events of CME. Therefore, we established a method for using Scanning Electron Microscopy (SEM) on metal replicas of unroofed *Arabidopsis* root protoplast cells, which allowed the direct examination of clathrin coated structures (CCSs). Availability of such a technique to study their size, shape and invagination enabled us to investigate the mechanism of CCV formation.

Metal replicas of *Arabidopsis* root protoplast cells provided a detailed view of well-preserved CCPs at various stages of invagination at the PM and fully formed CCVs deeper inside the cell, which are often attached to cytoskeletal and intracellular membrane structures (*Figure 1A* and *Figure 1—figure supplement 1A,B*). In addition to this apporach, we also used whole protoplasts resin embded sections and obsered the cross-sections of CCSs using TEM. However, this technique was less efficient than metal replica, as we could observe only a few CCSs and we could not obtain tomographic reconstructions to detemine the composition of clathrin polygons which make up the whole coat (*Figure 1—figure supplement 1C*).

Metal replica allowed us for the first time to observe clathrin triskelions assembling together to form the clathrin basket, making the vesicle coat. We proceeded to analyze the size and shape of the overall clathrin basket shape. This is because it provides insight into the heterogeneity of the arrangement and formations of the clathrin coat *in situ*, where multiple physiological factors can regulate the size and shape of CCVs (*Saleem et al., 2015*). By analyzing the surface-view of the entire

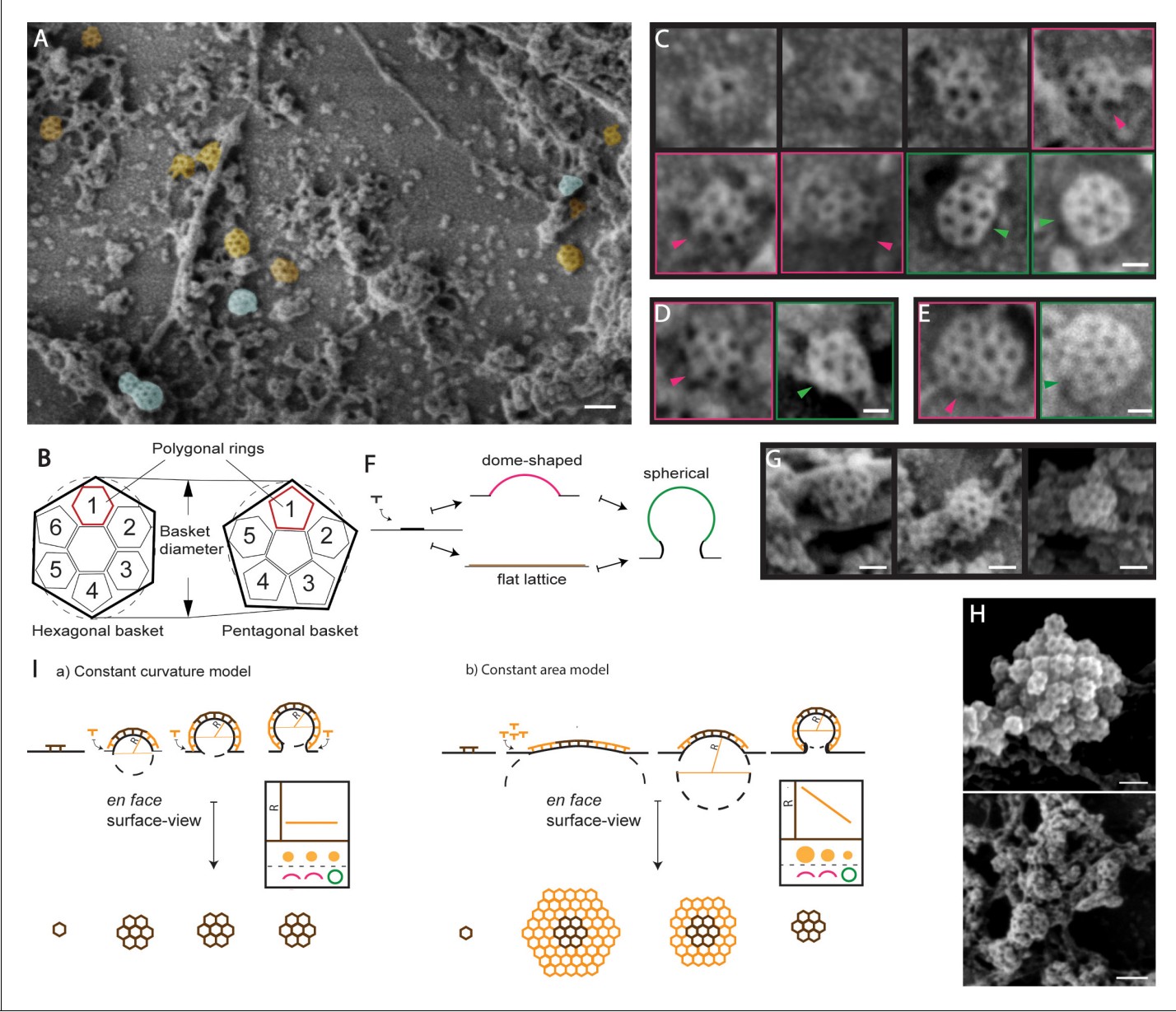

**Figure 1.** Ultrastructural characterization of clathrin-coated structures in unroofed protoplasts by SEM. (**A**) SEM image showing CCSs found associated to the PM (orange) and intracellular membrane structures (blue). See also *Figure 1—figure supplement 1A, B*. (**B**) Illustration of the *en face* surface-view of 'hexagonal' (left) and 'pentagonal' (right) clathrin basket (as outlined in black), which is composed of 6 and 5 polygonal rings respectively. (**C**) Representative images depicting various stages of development of the hexagonal basket-type CCV. (**D**) Representative pentagonal basket type CCSs (left and **E**) irregular basket type CCSs (right) at the PM. Color-coded boxing corresponds to the membrane invagination assessment illustrated in 1F; dome-shaped basket - pink, spherical basket - green. Arrows show the side-view of the baskets. (**F**) Membrane invagination assessment illustrating the side-view of CCSs with different degrees of invagination. See also *Figure 1—figure supplement 1H*. (**G**) Representative images of isolated CCSs of pentagonal, hexagonal and irregular basket types at the intracellular membrane structures. (**H**) Example aggregated and partial fused CCSs at the intracellular membrane structures. See also *Figure 1—figure supplement 1 E-G*. (**I**) Two models of clathrin-mediated membrane bending differentiated by *en face* surface-view analysis: The constant curvature model where, the added clathrin bends the membrane and the *en face* view of the CCS, depicting the radius of curvature (R), stays the same. The constant area model where the *en face* view of the CCS converges from a flat lattice to a curved dome and further to spherical, thus progressively decreasing R. The graph represents the relationship among R, *en face* surface-view and the degree of membrane invagination of the CCSs. Scale bars; 100 nm (A, E upper panel), 30 nm (**B, C**), 50 nm (**D**) and 60 nm (**E** lower panel). The online version of this article includes the following source data and figure supplement(s) for figure 1:

**Figure supplement 1.** Ultrastructural characterization of clathrin-coated structures.
**Figure supplement 1—source data 1.** Source data for the quantification in *Figure 1—figure supplement 1*.

clathrin baskets, rather than distinct polygonal clathrin rings (pentagonal, hexagonal or heptagonal rings) it is comprised of, we were able to categorize the CCSs into three distinct populations. The first characterized population consists of a clathrin basket with a central clathrin hexagonal ring surrounded by six other polygonal rings, where the final *en face* surface-view of the entire clathrin basket outlines a hexagonal structure, hence was categorized as hexagonal basket-type population (*Figure 1B* -left). The progression of this basket-type formation could be seen at the PM (*Figure 1C*). The second population of CCSs consists of a basket which has a pentagonal *en face* surface-view shape, with one central pentagonal ring surrounded by five other polygonal rings; hence termed pentagonal basket-type population (*Figure 1B* - right, 1D). In addition to these basket-type populations, a third was found. This population consists of an irregularly ordered basket with a larger diameter than the hexagonal and pentagonal basket-types. The hexagonal basket population has diameter, ranging from 46 to 81 nm with a mean of 66.1 nm. The diameter of pentagonal basket population range from 41 to 72 with a mean of 59.9 nm (15 cells, 188 CCSs measured). The diameter for the irregularly ordered baskets ranged between 62–113 nm with a mean of 85 nm (15 cells, 72 CCSs measured) (*Figure 1E*). A possible reason for the differences in sizes could be due differences in the numbers of hexagonal clathrin rings, as CCVs are expected to contain 12 pentagons and a different number of hexagons alters the overall size (*den Otter and Briels, 2011*; *Cheng et al., 2007*).

In addition to the PM, we examined the surface of intracellular membrane structures (*Figure 1A* and *Figure 1—figure supplement 1B*). We observed CCSs conforming to all typical basket types (*Figure 1G*), and the frequency distribution of the populations are similar in both the PM and the intracellular membrane; with the prominent type being hexagonal (60%) (*Figure 1—figure supplement 1D*). Typically for intracellular membranes there were multiple CCSs fused together, the occasional occurrence of huge CCSs and also large aggregations of CCSs comprising of same or different population types (*Figure 1H* and *Figure 1—figure supplement 1E–G*). These intracellular membrane structures have been previously reported to be the early endosome/trans-Golgi network (EE/TGN) or the partially-coated reticulum (*Galway et al., 1993*; *Kang et al., 2011*; *Robinson, 2015*; *Tanchak et al., 1988*). The CCSs at the intracellular membrane structures could be secretory vesicles budding off the TGN (*Kirchhausen, 2000*; *Kural et al., 2012*; *Robinson and Pimpl, 2014*; *Watanabe et al., 2014*) or if the uncoating in plants would be delayed, they can be also endocytosed CCVs fusing to the EE.

These observations show that metal replica of unroofed protoplast cells provide an unprecedented insight into various structural aspects of plant CME. It revealed the existence of distinct populations and arrangements of clathrin structures at the PM and various intracellular endosomal membrane structures.

## Constant curvature mode of clathrin-coated pit formation at the plasma membrane

Next, we used the metal replicas to determine the mechanism of CCV formation in plants. The mechanism of membrane bending has been studied extensively in yeast and mammalian models (*Kaksonen and Roux, 2018*; *Lampe et al., 2016*; *Sochacki and Taraska, 2019*), where two major models have been proposed (*Figure 1I*). The 'constant area' model; where clathrin initially forms a flat lattice on the PM, and then through continuous exchange of clathrin molecules bend the membrane. Thus, during membrane bending, the radius of curvature continuously decreases (*Avinoam et al., 2015*; *Kaksonen and Roux, 2018*; *Scott et al., 2018*). Moreover, in mammalian systems CCPs also mature and bud off the large clathrin-coated surfaces: plaques and flat clathrin lattices (*Grove et al., 2014*; *Lampe et al., 2016*; *Leyton-Puig et al., 2017*). The second model is the 'constant curvature' model. The clathrin assembly at the invagination directly bends the membrane and the continuous polymerization develops the curvature. Such a mode of development maintains the radius of curvature (*Bucher et al., 2018*; *Kaksonen and Roux, 2018*; *Lampe et al., 2016*; *Scott et al., 2018*). In order to further explore the mechanism of membrane bending in plant system, we used the SEM images of unroofed protoplast cells, since different stages of CCP development were evident.

First, we found that there was an absence of large clathrin assemblies or lattices in *Arabidopsis* (*Figure 1A* and *Figure 1—figure supplement 1A*). Most of the lattices that were found were small, with only a few clathrin polygons, consistent with early reports in other plant species (*Coleman et al., 1988*; *Emons, 1986*).

To differentiate between the two main models of membrane bending, we developed a *en face* surface-view analysis of CCSs using SEM images of the metal replicas of protoplast cells. First we classify the degree of membrane invagination by assessing the 'side-view' of the CCSs, which informed if the CCS is (i) a flat lattice, or (ii) dome-shaped and developing attached to PM, or (iii) spherical and mature (*Figure 1F*). Then by analyzing the *en face* surface-view of the CCV, we can assess the shape of the entire basket of the flat, dome-shaped and spherical CCPs. The *en face* shape of the basket of CCSs can been used as a proxy for the radius of curvature. If the *en face* view of the basket changes from a large flat clathrin lattice to a smaller dome-shaped to a much smaller spherical structure with decreasing radii of curvature, it is indicative that the membrane bending follows constant surface model. If the *en face* view of the basket (hence the radius of curvature) stays the same along the dome-shaped and the spherical CCSs, it is indicative that the membrane bending follows the constant curvature model (*Figure 1H*).

We therefore classified all the PM associated CCPs (151 in total) in our protoplast cell replica SEM images into flat, dome-shaped and spherical CCPs (*Figure 1—figure supplement 1H*). We found that less than 4% of the CCPs formed a flat structure (*Supplementary file 1* - table 1). We then analyzed the *en face* surface-view of the dome-shaped and spherical CCPs and found that 47 of the dome-shaped and 16 of the spherical CCPs exhibited the same *en face* view; the hexagonal basket shape. This result highlights that hexagonal basket-type CCPs at different stages of development possess a constant radius of curvature (*Figure 1C and H* and *Supplementary file 1* - table 1). We also observed a similar result in CCPs, which had a pentagonal basket-type (*Figure 1D* and *Supplementary file 1* - table 1). This demonstrates that predominantly the CCPs constituting the hexagonal and pentagonal basket-type populations undergo membrane bending following the constant curvature model.

However, we could not entirely dismiss the existence of other mechanism of membrane bending, given that there was a small percentage of flat CCSs that could potentially develop and bend to form a spherical CCV, characteristic of constant area model. In addition, due to the irregular ordering of the clathrin in the irregular basket-type CCVs (*Figure 1E*), it is impossible to extrapolate lattice rearrangement and membrane folding mechanism of the irregular population of CCSs.

Interestingly, the diameters of the CCVs measured in our unroofed protoplast cells is similar to CCVs purified from the entire seedlings (where the CCV size ranges between 60–80 nm [*Mosesso et al., 2018*; *Reynolds et al., 2014*]). Therefore, it is very likely that the both the protoplasts and intact plant systems have similar mechanisms of CCV formation. To further determine the membrane bending mechanism in the root system, we used live imaging with TIRF-M, which excels at looking at cell surface processes due to its shallow illumination penetration depth of ~100 nm (*Axelrod, 2001*). We directly observed the PMs of root epidermis expressing clathrin light chain two tagged with GFP (CLC2-GFP) and detected small foci rather than large plaques of fluorescence signal (*Figure 2A*). There are large clathrin structures present in the live tissue images, however they are highly mobile (*Figure 4—video 2* – control condition, and *Figure 5—video 3*) and represent EE/TGN structures (*Figure 1F*), therefore they are not attached to the PM. We also generated the intensity time course of the fluorescent signal obtained from clathrin tracks. The intensity profile show an initial steep increase of signal (representing the assembly of clathrin at the pit), followed by a plateau phase (representing the maturation phase of the CCP), and finally a sharp drop in intensity (corresponding to the scission and release of the CCV) (*Figure 2B*); which is reminiscent of the clathrin intensity profiles reported in mammalian cells (*Loerke et al., 2009*). The maturation phase accounts for only 17.8 ± 8% of the entire span of development, while the assembly phase takes up to 42 ± 5%. Such segregation in the intensity profile is suggestive of the 'constant curvature', rather than 'constant area' model, which entails a short assembly and very long maturation phases (*Avinoam et al., 2015*).

In summary, ultrastructural and live imaging analysis of clathrin at the PM did not detect any larger clathrin lattices that could develop into vesicles. Moreover, the ultrastructure observations of developmental stages of CCPs in root protoplasts suggest that the membrane bending follows 'constant curvature model' supported also by live imaging-based clathrin intensity profiles in *Arabidopsis* roots.

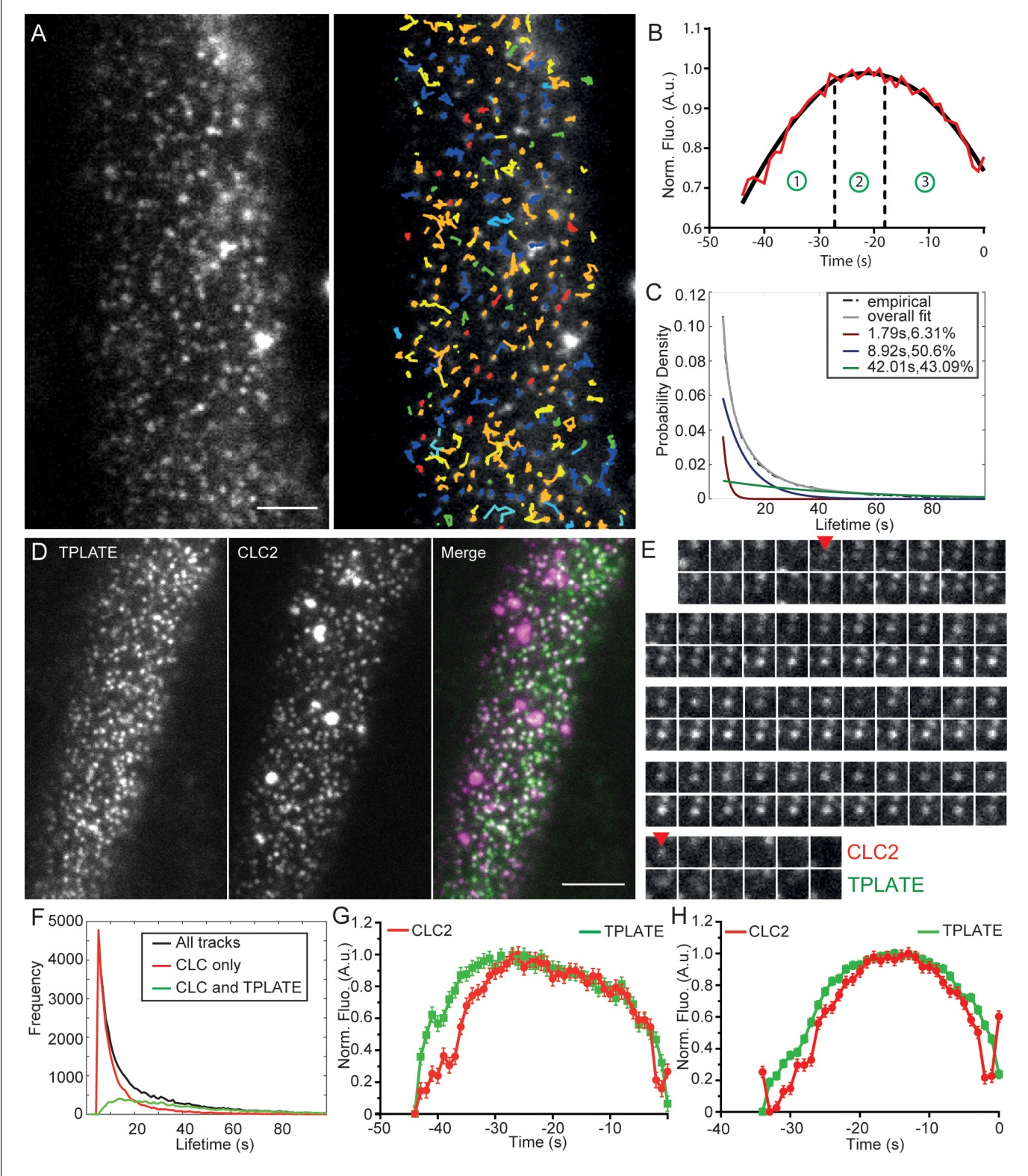

**Figure 2.** Characterization of clathrin kinetics at the cell surface. (A) TIRF-M image of clathrin foci in the root epidermal cell expressing CLC2-GFP (left) and the automated tracking results (right). (B) Intensity-time course of the combined mean fluorescence profile (along with a smoothening curve) of CLC2-GFP in root epidermal cells. The intensity profile depicts different CCP developmental phases; (1) assembly, (2) maturation and (3) scission. The points represent mean ± SEM of the mean intensity of all the trajectories that have the mean lifetimes. N = 5 cells from individual roots, 182 trajectories. (C) Normalized histogram of CLC2-GFP tracks (black) with three exponential distributions fitted (red, blue and green). The inset denotes the mean

*Figure 2 continued on next page*

*Figure 2 continued*

lifetime and the percentage contribution to the total population of each sub population. N = 12 cells from individual roots, 149162 tracks. Also see *Figure 2—figure supplement 2*. (D) Representative dual channel TIRF-M image of root epidermal cell expressing TPLATE-GFP and CLC2-RFP. (E) Time series of an isolated endocytic event positive for CLC2-RFP and TPLATE-GFP. Arrows mark the appearance and the disappearance of CLC2. Quantification of this event is shown in figure S4. (F) Frequency distribution of the lifetimes of clathrin and TPLATE tracks. The inset indicates the color codes marking the frequency distributions of different sets of tracks. N = 6 cells from individual roots, 45337 tracks. (G) Clathrin departure assay of TPLATE-GFP and CLC-tagRFP in root epidermal cells. N = 6 cells from individual roots, 16756 tracks. (H) Clathrin departure assay of TPLATE-GFP and CLC-tagRFP in hypocotyl. N = 3 cells from individual hypocotyls, 1460 tracks. Scale bars; 10 µm (A and D).

The online version of this article includes the following source data and figure supplement(s) for figure 2:

**Source data 1.** Source data and code for the quantification for *Figure 2*.
**Figure supplement 1.** Co-localization of CHC1 and CLC2.
**Figure supplement 2.** Goodness of fit analysis of clathrin sub-populations.
**Figure supplement 3.** Example TPLATE and CLC2 positive events.

## Productive populations of clathrin-mediated endocytic events at cell surface

While TIRF-M has been used in plants (*Johnson and Vert, 2017*; *Vizcay-Barrena et al., 2011*; *Wan et al., 2011*), a detailed analysis of the clathrin kinetics and characterization of endocytic events at the cell surface has not yet been performed. With improvements of imaging protocols, automated particle detection and tracking analysis is now feasible (*Johnson and Vert, 2017*; *Wang et al., 2015*). We therefore combined TIRF-M with automated particle detection and tracking analysis to characterize in detail the CME kinetics on the cell surface.

A large data set of lifetimes was generated (12 cells and 149162 tracks) by imaging plant root epidermis expressing CLC2-GFP. CLC2 was used as a marker for CME, as it is one of the key components of the clathrin coat, comprised of both clathrin light and heavy chains. Additionally, we found a high degree of co-localization (~62%) between CLC2 and CHC1, indicating that the CLC2 is present at the majority of total CME events (*Figure 2—figure supplement 1*). The mean lifetime of all the clathrin CLC2 tracks was 20 s, which closely matches the lifetime reported based on manual measurements (*Konopka et al., 2008*). The histogram of this data set appears to be exponential (*Figure 2B*), suggesting that the majority of clathrin imaged were in fact transient events. Therefore, to further characterize the kinetic behavior of clathrin, and uncover the lifetime of the *bone fide* clathrin endocytic population, we used an approach based on a mixture model distribution fitting (*Loerke et al., 2009*). Further statistical testing of the experimental data revealed that three populations produced the best fit (*Figure 2—figure supplement 2*). There were two short-lived sub-populations identified with lifetimes of 2 s (accounting for 6.31% of the total population) and 9 s (accounting for 50.6% of the total population). The third population accounts for 43.9% of all clathrin on the surface and had a lifetime of 42 s (*Figure 2C*).

To confirm if this longer-lived sub-population represents *bona fide* CME, we used dual channel TIRF-M on root cells expressing CLC2-GFP and an additional endocytosis marker TPLATE, which has been proposed to function as an endocytic adaptor (*Gadeyne et al., 2014*). One would expect *bona fide* CME events to be positive for both markers and indeed, the mean lifetime of the tracks positive for both markers was 42 s (*Figure 2C,D,E* and *Figure 2—figure supplement 3*). We also conducted clathrin departure assays on these dual channel experiments. Briefly, the end of the clathrin track depicts when it has left the illumination area of the TIRF evanescent wave, thus representing when the vesicle has been scissioned from the membrane (*Johnson and Vert, 2017*; *Mattheyses et al., 2011*; *Merrifield et al., 2002*). Tracks with the mean lifetime were combined to generate a mean fluorescent intensity profile of both channels relative to the departure of clathrin. TPLATE shows concurrent recruitment to the site of endocytosis, and clathrin shows a profile which fits the expected intensity profile of CME event; where it gradually assembles as the invagination grows (assembly phase), plateau (maturation phase) and then decreases upon scission and release (*Figure 2G*). Additionally, the lifetime of CLC2 when colocalized with another adaptor AP2 (*Di Rubbo et al., 2013*) confirmed these observations, as its lifetime was found to be similar, 43 s. Thus, the empirically obtained *bona fide* population of endocytic events matches the mathematically simulated longer-lived sub-population collectively suggesting the lifetime of CME events in root to be about 42 s.

Use of the intact plants allowed us to compare the kinetics of CME in different organs. The same analysis of CLC2 and TPLATE positive tracks in *Arabidopsis* hypocotyl epidermis produced a significantly shorter mean lifetime of 33 s (*Figure 2H*).

Thus, the mathematical analysis of CLC2 lifetime histograms and dual channel TIRF-M imaging independently reveal a lifetime of about 42 and 33 s for the *bona fide* CME events in root and shoot cells, respectively and suggested a so far unappreciated organ-specific regulation of CME in plants.

## Absence of actin accumulation at the endocytic foci

Clathrin pit invagination requires the use of accessory proteins to provide force, in order to overcome the energetic cost of physically altering the membrane structure against the resistive forces of the cell. In yeast, and in mammalian cells with high membrane tension, actin is considered the major force generator aiding membrane invagination (*Kaksonen and Roux, 2018*; *Kaksonen et al., 2005*; *Saleem et al., 2015*; *Stachowiak et al., 2013*). Actin polymerizes around the endocytic pit and with pushing and pulling actions, it aids the invagination to form against the tension and turgor pressure (*Brady et al., 2010*; *Buser and Drubin, 2013*; *Picco et al., 2015*; *Toshima et al., 2006*). Plant cells, similar to yeast are enclosed within a cell wall and possess high turgor pressure values 0.3–1 MPa (*Cosgrove, 1993*); therefore actin has been assumed to play a key role in membrane invagination during plant endocytosis (*Chen et al., 2011*; *Robatzek et al., 2006*).

Using SEM on unroofed protoplasts, we were able to observe the arrangement of cortical actin. Deeper inside the cell, actin exhibits a more complicated network (*Figure 3—figure supplement 1A*) but closer to the PM, we observed only thick main filaments and several thinner subsidiary filaments but, unexpectedly, never an enrichment of actin around the CCPs (*Figure 3A*).

In complementary experiments, we analyzed transgenic plants expressing Actin-Binding Domain 2-GFP (ABD2-GFP) and CLC2-mKO, using TIRF-M. We were able to observe the intact cortical actin filaments of hypocotyl epidermal cells (*Figure 3B*), which formed a filamentous network of cortical actin consisting of thick main filament and numerous branches of dynamic subsidiary filaments, as previously reported (*Staiger et al., 2009*). In accordance with our SEM analysis of the protoplast replicas, we observed no actin concentrations at the PM around the CCPs marked by CLC2-mKusabira-Orange (mKO). In addition, in root hair cells, when we assessed the localization of actin and the localizations of CCPs at the PM (marked by Dynamin-Related Protein (Drp1C)), actin did not form concentrate foci on the PM, unlike the CCP markers (*Figure 3C*). Examination of additional actin markers (Lifeact, ABD2 and mTalin) also failed to reveal accumulations of actin around sites of CME (*Figure 3—figure supplement 1B*) .

Thus, in contrast to expectations based on observations in yeast and some mammalian systems, where CCP invagination is facilitated by actin polymerization, plant cortical actin only forms filaments and does not accumulate around the invaginating CCPs as concrete foci.

## Actin is not mandatory during CCP formation and cargo internalization

The absence of actin foci at the endocytic spots is unexpected given that actin polymerization energy had been proposed to be a critical factor for plant CCP development.

In order to test the importance of actin in CCV formation, we perturbed actin by either stabilization of actin filaments treating with Jasplakinolide (Jasp) (*Figure 3—video 1*) or depolymerisation (*Figure 3—figure supplement 1C* and *Figure 3—video 2*) treating with LatrunculinB (LatB). Using TIRF-M, we directly visualized the dynamics of the endocytic events at the PM using different endocytic markers; TPLATE, CLC2 and DRP1C. After prolonged treatment with 10 µM Jasp that lead to extreme structural abnormalities of the actin filaments, or 10 uM LatB that lead to overall depolymerisation of the actin network, CCVs marked by CLC2-mKO formed and were removed from the PM with no apparent arrest of pits (*Figure 3—videos 1* and *2*). Moreover, we observed no clear changes in the overall lifetime distribution and foci density of any of these EAPs after 10 µM LatB and 5 µM Jasp treatments in both hypocotyl and root epidermal cells (*Figure 3D* and *Supplementary file 1* - table 2).

We also examined whether actin perturbations cause disturbances in CCP development, such as delays in assembly and/or maturation. We observed the intensity profile of *bona fide* endocytic subpopulation of CCPs with the lifetime of 44 s (*Figure 3E*) and sub-populations with lifetime of 22 s, the overall mean of the total population (*Figure 3—figure supplement 1D*). Again, we saw no

significant changes in the average duration of any of the developmental phases: CCP initiation, maturation or scission, after LatB treatment. All these results show that depolymerizing or stabilizing actin has no effect on CCP development and the rate of endocytosis.

Next, we tested the importance of intact actin on CCP productivity by observing the internalization of different endocytic cargoes. We used the endocytic tracer, the dye FM4-64 (*Jelínková et al., 2010*) in root cells and observed that after LatB and Jasp treatments, PM internalization still occurred (*Figure 3F* and *Figure 3—figure supplement 1E*), consistent with observations in a more primitive plant *Chara* (*Klima and Foissner, 2008*). We also tested the effect of actin perturbation on receptor-mediated endocytosis in hypocotyls by following flagellin (flg22)-mediated internalization of its receptor FLS2. After flg22 application, FLS2 is endocytosed and can be visualized in ARA7-labeled endosomes after 40 min (*Beck et al., 2012*). Again, following LatB treatment, FLS2 endocytosis still occurred and the receptors reached the endosomes, as expected (*Figure 3G*). Next, we directly measured the rate of internalization of PIN2; a PM-localized auxin transporter in root epidermal cells, which undergoes constitutive endocytosis and recycling to the PM (*Adamowski and Friml, 2015*). PIN2 was tagged with photo-convertible Dendra and the existing PIN2 population in the root was photo-converted to red leaving the *de novo* synthesized population green. The loss of photo-converted red PIN2 PM signal was monitored over time, which denotes the rate of its endocytosis.

To validate this assay, we examined the PIN2-Dendra endocytic rate by overexpressing AUXILIN-LIKE2, which has been shown to inhibit CME (*Adamowski et al., 2018*), and found that the PIN2 endocytic rate was reduced (*Figure 3H*). In contrast, LatB treatment led to no significant change in the PIN2 endocytic uptake (*Figure 3H*). These results show that the intact actin is not required for CME of multiple cargoes.

In conclusion, these multiple quantitative observations show that acute as well as chronic perturbations of the actin network do not have a major impact on CCP formation, CME dynamics or CME of multiple cargos in both *Arabidopsis* roots and hypocotyls.

## The role of actin in EE/TGN dynamics and efficient endocytic trafficking

Next, we investigated a possible role of actin in post-endocytic events, such as vesicular trafficking and EE formation. In contrast to mammalian cells, which typically utilize microtubules for subcellular organelle trafficking, plant cells utilize actin filaments (*Breuer et al., 2017*; *Geldner et al., 2001*; *Granger et al., 2014*).

Each cell has several Golgi apparatus (GA) fused with EE/TGN compartments; which is specific to plant systems (*Kang et al., 2011*; *Viotti et al., 2010*). The EE/TGN compartments continuously form at, mature and detach from the trans-side of the GA (*Kang et al., 2011*) and dynamically move along actin filaments (*Figure 4—video 1*). Following depolymerizing or stabilizing the actin filaments, we observed a complete halt in the dynamics of the GA and the EE/TGN, which resulted in local aggregations of EE/TGNs, as shown by imaging epidermal cells in hypocotyl (*Figure 4A*; *Figure 4B* and *Figure 4—video 2*). Despite this loss of dynamics, and the resulting aggregations also in root epidermal cells, we observed that FM4-64 was still able to reach the agglomerated EE/TGN (*Figure 4C*). Nonetheless, we observed pronounced defects in the post-endocytic trafficking of cargoes such as PIN2. PIN2-GFP, which is normally observed at late endosomes marked by ARA7, was strongly mislocalized after prolonged actin perturbation. This was manifested by abnormally high amount of PIN2 in the aggregated late endosomal structures in root cells (*Figure 4D*). These trafficking defects are in agreement with our previous observation where interference of actin affected the arrival of PIN2 to the vacuole (*Kleine-Vehn et al., 2008*). This suggests that actin is required for the GA and EE/TGN dynamics and for the overall efficiency of post-endocytic trafficking processes.

## Actin-mediated early endosome dynamics for collecting and guiding CCVs

Next, we examined the actin dynamics in relation to the endocytic events at the PM in detail. Using TIRF-M, in hypocotyl epidermis, we observed the dynamics of actin subsidiary filaments (marked by ABD2-GFP) and the CCVs (marked by CLC2-mKO). We observed a spatial and temporal correlation between the appearance of a small branch of an actin subsidiary filament and the disappearance of CCVs from the PM (*Video 1*). This could possibly mean that the CCVs, after the scission and release,

get whipped away by actin subsidiary filaments to get them moving along the network of actin. This live dynamic interaction is further supported by evidence from replicas of protoplasts where fully formed CCVs were observed attached along actin filament in close proximity to intracellular endosomal membrane structures (*Figure 5C*). To gain a further insight into the possible role of actin in organizing the early post-endocytic trafficking, we followed the actin-mediated dynamics of the GA and the EE/TGN along with the CCVs. The EE/TGN moves typically synchronously with GA, loosely associated with it (*Kang et al., 2011*). While studying movement of these structures, we observed that on their way below the PM, they pick up/collect the scissioned CCVs (*Figure 5A*, *Figure 5—videos 1* and *2*). To provide an unbiased verification of this phenomenon, we developed an automated 'endosomal pick-up' analysis in root and hypocotyl cells. This provided a percentage of productive CCV tracks (identified by being positive for both CLC and TPLATE) that ended in the 'endosomal' regions of interest (ROIs) corresponding to CCVs being scissioned while the endosome was passing close by (*Figure 5—video 3*). We found that there was a significant increase in the percentage of CCV tracks ending in the 'endosomal' ROIs compared to 'control' ROIs (*Figure 5B*) providing unbiased and quantitative evidence that actin-organized movement of EEs aids collecting the CCVs after their scission.

All these observations together with the *in vivo* imaging of endocytotic events at the PM suggest that actin filaments traffic the CCVs directly from the site of endocytosis. Also, actin guides the EE/TGN compartments movement to 'collect' CCVs as they move. This is not only a first observation how EE in plants are formed by directional traffic and fusion of CCVs, but it also suggests a more active role of endocytic compartments in organizing the early post-endocytic events (*Fürthauer and González-Gaitán, 2009*; *González-Gaitán, 2003*; *Platta and Stenmark, 2011*).

## Delayed and sequential uncoating of scissioned clathrin-coated vesicles

Many well established models of CME assume that CCVs shed their coat rapidly after the scission event, in order to be able to fuse with EE (*Beck et al., 1992*; *Böcking et al., 2011*; *Massol et al., 2006*; *Sekiya-Kawasaki et al., 2003*). The key mammalian uncoating proteins, auxilins and Heat shock cognate 71 kDa protein (HSC70) proteins, have been shown to be crucial for efficient endocytosis (*Bai et al., 2010*; *Greener et al., 2000*; *Hirst et al., 2008*; *Lee et al., 2008*; *Yim et al., 2010*). While auxilins appear to be conserved in plants, they are not essential for endocytosis or development (*Adamowski et al., 2018*), implying that uncoating in plants may work differently.

While closely following the CCVs and the EEs with TIRF-M in hypocotyl epidermis, we noticed that while some CCVs lose their clathrin coat (*Video 2*), many others retained their coat all the way till they reached the EE/TGN compartment (*Figure 5A*, *Figure 5—videos 1* and *2*). This implies that the clathrin coat does not get disassembled immediately after scission, unlike in animal cells.

Following this unexpected observation, we analyzed the mechanism of uncoating in further detail. We closely observed the clathrin-coat disassembly by following two components of CCVs; the coat protein, CLC2 and an adaptor protein, TPLATE. While tracking CCVs leaving the PM, we observed CCVs that immediately fuse into the EE and CCVs that are present in the cytosol for a longer time period. For the CCVs immediately fusing with EEs, both clathrin coat and adaptor were retained until the vesicles reached the EE. However, after fusion, the EE compartment did not contain any visible adaptor, implying that the uncoating process had taken place, at least partially, but immediately before or during the fusion (*Figure 6B* and *Figure 6—video 1*). This hypothesis is supported also by ultrastructural analysis, where aggregates of partially coated CCVs fused to the intracellular endosomal membrane were detected (*Figure 1H*). This also supports that some CCVs remain partially uncoated at the EE.

For the CCVs being trafficked for a longer duration in the cytosol before reaching the EE, we were often able to observe the loss of adaptor before clathrin coat (*Figure 6A* and *Figure 6—video 2*). This means that the disassembly of the released CCVs does not typically happen all at once but sequentially. The sequential loss of coat was observed in the majority of cases (18/20 CCVs); whereas in the remaining traced CCVs both components vanished together. This sequential uncoating is also supported by TEM observations, in which we can detect partially uncoated CCVs deeper in the cytosol (*Figure 6C*).

In summary, multiple independent observations show that in plants uncoating of the CCVs does not happen immediately (co-operative process) after scission at the endocytic spot, but the clathrin

coat is retained for a prolonged time, shedding its components gradually (sequential process) on its way to the EE (*Figure 6D*).

## Discussion

Endocytosis is a crucial cellular process in all eukaryotes regulating a multitude of fundamental cellular functions. In plants, CME has emerged as the major, if not the exclusive (*Dhonukshe et al., 2007*), mode of endocytosis crucial for number of developmental and physiological processes. Nonetheless, a precise characterization of plant CME is greatly lacking when compared to other model systems. By developing and advancing imaging methods, we were able to provide a detailed characterisation of plant CME in *Arabidopsis* tissues. We found that while there is some evolutionary conservation of endocytic molecular components between plants and other model systems (*Baisa et al., 2013*; *Chen et al., 2011*), mechanism of plant CME and post-endocytic processes shows multitude of unique features compared to other model systems.

### Key characteristics of plant CME

A major reason for the lack of plant CME characterization is poor implementation of techniques to directly examine CCVs and live CME events in plant cells. Here, by combining advanced SEM and TEM, advanced live imaging and automated, quantitative analysis techniques, we provide the detailed visualization of every stage of the CCV formation and trafficking away from the PM to the EEs, at unprecedent resolutions.

   We determined that the typical *Arabidopsis* CCVs possess a hexagonal clathrin basket coat, with an average diameter of 60 nm. Further work is needed to optimise protocols to determine the exact arrangements of individual clathrin polygons and what determines their arrangements. They are located on the PM, on actin filaments and at aggregations at the EEs/TGNs. Combining direct ultra-structural analysis of CCPs with high temporal resolution live imaging of CLC2 allowed us to determine that *Arabidopsis* CCP formation predominantly follows the 'constant curvature model'. Moreover, the presence of small flat lattices, which could give rise to CCVs following 'constant area model' or recently described 'intermediate model' (*Bucher et al., 2018*; *Scott et al., 2018*), suggests the existence of more than one mechanism of membrane bending within the plant system.

   The large-scale examination of live *Arabidopsis* revealed the existence of 3 kinetically distinct populations of CLC2 on the root PM. While the majority of the events were in fact transient and likely non-functional, the *bona fide* CME population had a mean lifetime of 42 s and accounted for 44% of the total CME events at the PM. The discovery of differences in the *bona fide* lifetime between tissues suggest that there are organ- and tissue-specific modulations of CME in plants.

   Compared to yeast models (*Lu et al., 2016*), the plant CME appears to be much faster (considering all phases of CCV formation: assembly, maturation and scission) and the resulting endocytic vesicles are almost twice the size (*Kaksonen and Roux, 2018*). This suggests that plants have differential regulatory processing of recycling clathrin compared to animals and yeast.

   It is important to note that our live imaging used CLC2 as a marker for CME, despite biochemical and genetic evidence that the CLCs interact with both CHC1 and CHC2 (*Adamowski et al., 2018*; *Gadeyne et al., 2014*) and the CHCs being genetically redundant for each other (*Kitakura et al., 2011*), we found a non-perfect localisation of CLC2 and CHC1. Therefore, further work is required to determine if the other isoforms of clathrin share the dynamics *in planta*.

### Actin-independent mechanism of plant CME

Actin has been emphasized to be important for CME in mammals and crucial in yeast, where it is required for detaching the PM from the cell wall, and for deforming the membrane locally against the turgor (*Leyton-Puig et al., 2017*; *Tweten et al., 2017*). In further support of this, yeast and mammalian systems also have acto-myosin network strongly accumulated at the endocytic spot (*Collins et al., 2011*; *Picco et al., 2015*; *Toshima et al., 2006*; *Yarar et al., 2005*). Therefore, actin has long been assumed to be a critical component of plant CME (*Chen et al., 2011*; *Robatzek et al., 2006*; *Samaj et al., 2004*). However, chronic and acute inhibition of actin dynamics demonstrated that actin perturbations have no visible influence on CME kinetics or efficiency. Furthermore, plants do not possess type one myosin (the class of myosin involved in yeast endocytosis), and, more importantly, actin or its associated proteins do not accumulate at the CME loci. This

challenges all models that frame our current understanding on membrane cycling energetics under turgor.

The mechanism of plant CME membrane bending was found to follow the constant curvature model. In this model, the curvature could be generated by the continuous polymerization of clathrin (*Bucher et al., 2018*; *Scott et al., 2018*), therefore in plants it is possible that clathrin polymerization energy, in addition to other membrane deforming proteins, might provide enough force to overcome the turgor pressure. It is noteworthy that plant-specific mechanism of membrane bending must overcome turgor pressures higher than found in yeast, while producing larger CCVs (60 nm compared to 36 nm in *Saccharomyces cerevisiae Smaczynska-de Rooij et al., 2010*) and in a more rapid manner (33–42 s compared to 75–135 s *Lu et al., 2016*).

It is possible that during the independent evolution of multi-cellularity in plants and animals, alternate membrane deformation mechanisms co-evolved in accordance to the differences in cell-cell adhesion, signalling and polarity between the model systems (*Eaton and Martin-Belmonte, 2014*). Therefore, screening for other membrane modifying proteins and studying their activity would enable us to discover this plant specific substitution mechanism for membrane deformation during CME.

## Active role of actin-mediated endosomal movement in endocytosis

Our observations established that while actin does not accumulate at sites of CME and it is not required for CME events at the PM, it mediates directional movement of CCVs and EEs. However, during actin disruption, endocytosed membrane still reached the aggregated EE/TGN structures. This suggests that a passive Brownian motion of the freed CCVs is sufficient for the eventual reaching of CCVs the EEs.

However, direct examination of protein cargos and their mis-localization following actin disruption suggested that actin-mediated EE mobility is a key requirement for a specific cargo sorting. In support of this, we observed a strong spatio-temporal relationship between the actin filaments dynamic at the vicinity of the PM and CME events (*Figure 7*). Further to this, we also observed a significant correlation between the termination of CME events at the PM and the targeted movement of the EE/TGN passing below (*Figure 7*).

This suggests a so far unappreciated active role of fine actin filament dynamics as well as the directional, actin-mediated TGN/EE movement in mediating CME events at the PM and organizing early post-endocytic trafficking.

## Delayed uncoating of clathrin-coated vesicles on route to endosomes

In both yeast and animal cellular systems endocytic vesicles uncoat rapidly after the scission event with the aid of auxillin and HSC70 proteins (*Böcking et al., 2011*; *Krantz et al., 2013*). However, there is some debate as to whether uncoating is an all-or-nothing process (*Schroeter et al., 2016*; *Trahey and Hay, 2010*). The uncoating process in plants has not been studied in any detail; while there are auxillin-like proteins, their deletion does not produce as strong effects on CME as observed in mammalian systems (*Adamowski et al., 2018*). This raises a possibility that plants have different mechanisms to uncoat the CCVs or uncoating is less crucial than assumed.

Our *in planta* live imaging show that CCVs had a prolonged retention of their clathrin coat, sometimes even until CCVs reach the EE (*Figure 7*). This opens a large scope for possible coat functions in CCV – EE fusions. Notably, not all constituents of the CCV coat were retained at the same time, as demonstrated by observations of CCVs that lost the plant-specific EAP TPLATE, while still retaining the clathrin coat. Such a sequential removal of CCV coat components is concordant to the Ark1-Prk1-synaptojanin mediated uncoating of CCVs observed in yeast (*Boettner et al., 2012*; *Sekiya-Kawasaki et al., 2003*; *Toret et al., 2008*). But retaining the clathrin coat till the CCVs fusion with the early endosomes is unheard of in other systems.

In summary, these observations revealed sequential and severely delayed uncoating or even no loss of clathrin coat on the way of endocytic vesicles to the early endosomes, which is in stark contrast with current models of CME.

## Conclusions

Here we present a detailed investigation into the mechanisms of CME progression from the initial invagination formation to its fusion with the endosomes thereby greatly advancing our current knowledge of the mechanism of plant CME. Plant CME appears to follow the constant curvature model of CCV formation occurring much faster and generating much larger vesicles than yeast systems (*Kaksonen and Roux, 2018*), of predominantly a hexagonal basket type. Contrary to current paradigm, plant CME does not require actin suggesting that plants have evolved a unique mechanism of membrane bending against turgor. Actin, nonetheless, is important in post-endocytoic events for movement of EEs and proper sorting of cargos. Importantly, the presence of dynamic fine actin filaments and passing of mobile EE below the PM has a significant correlation with an increased rate of CME, thus highlighting an unsuspected role of actin-mediated endosome movement in mediating CME at the PM and 'collecting' the post-endocytic vesicles. The strongly delayed uncoating and sequential removal of coat components is another unique feature of plant CME. These observations in complex, multi-cellular organs also provide unprecedented insight into higher order interactions, which are otherwise absent in unicellular models. Altogether, this study shows that despite the presence of evolutionary conserved components, plant CME is mechanistically unique, presumably reflecting an independent evolution of multicellularity in plants versus other models.

# Materials and methods

**Key resources table**

| Reagent type (species) or resource | Designation | Source or reference | Identifiers | Additional information |
|---|---|---|---|---|
| Cell line (*Arabidopsis thaliana*, Col-0) | Suspension cultured root protoplasts | | | |
| Biological sample (*Arabidopsis thaliana*, Col-0) | Seedlings | | | |
| Gene (*Arabidopsis thaliana*) | AP2A1 | The Arabidopsis Information Resource | AT5G22770 | |
| Gene (*Arabidopsis thaliana*) | Axl2 | The Arabidopsis Information Resource | AT4G12770 | |
| Gene (*Arabidopsis thaliana*) | CHC1 | The Arabidopsis Information Resource | AT3G11130 | |
| Gene (*Arabidopsis thaliana*) | CLC2 | The Arabidopsis Information Resource | AT2G40060 | |
| Gene (*Arabidopsis thaliana*) | Fim1 | The Arabidopsis Information Resource | AT4G26700 | |
| Gene (*Arabidopsis thaliana*) | FLS2 | The Arabidopsis Information Resource | AT5G46330 | |
| Gene (*Arabidopsis thaliana*) | PIN2 | The Arabidopsis Information Resource | AT5G57090 | |
| Gene (*Arabidopsis thaliana*) | Tplate | The Arabidopsis Information Resource | AT3G01780 | |
| Gene (*Arabidopsis thaliana*) | VHA-a1 | The Arabidopsis Information Resource | At2g28520 | |
| Genetic reagent (*Arabidopsis thaliana*) | pCLC2::CLC2-GFP | (*Konopka et al., 2008*) | | |
| Genetic reagent (*Arabidopsis thaliana*) | pLAT52::TPLATE-GFP x pRPS5::CLC2-RFP | (*Gadeyne et al., 2014*) | | |
| Genetic reagent (*Arabidopsis thaliana*) | 35S::ABD2-GFP | (*Kost et al., 1998*) | | |
| Genetic reagent (*Arabidopsis thaliana*) | p35S::CLC2-mKO | (*Naramoto et al., 2010*) | | |

*Continued on next page*

*Continued*

| Reagent type (species) or resource | Designation | Source or reference | Identifiers | Additional information |
|---|---|---|---|---|
| Genetic reagent (*Arabidopsis thaliana*) | pFLS2::FLS2-GFP x pUBQ10::mRFP-ARA7 | (*Beck et al., 2012*) | | |
| Genetic reagent (*Arabidopsis thaliana*) | XVE >> Axl2 x pPIN2::PIN2-Dendra | (*Adamowski et al., 2018*) | | |
| Genetic reagent (*Arabidopsis thaliana*) | pVHA-a1::VHA-a1-GFP | (*Dettmer et al., 2006*) | | |
| Genetic reagent (*Arabidopsis thaliana*) | pPIN2::PIN2-GFP | (*Xu and Scheres, 2005*) | | |
| Genetic reagent (*Arabidopsis thaliana*) | p35S::ARA7-mRFP | (*Ueda et al., 2004*) | | |
| Genetic reagent (*Arabidopsis thaliana*) | p35S::ST-RFP x pCLC2::CLC2-GFP | (*Ito et al., 2012*) | | |
| Chemical compound | LatrunculinB | Sigma Aldrich | L5288 | |
| Chemical compound | Jasplakinolide | Santa Cruz Biotechnology | sc-202191 | |
| Chemical compound | FM4-64 | ThermoFisher Scientific | T3166 | |

## Contact for reagent and resource sharing

Further information and requests for resources and reagents should be directed to and will be fulfilled by the Lead Contact, Jiri Friml (jiri.friml@ist.ac.at).

## Experimental model and subject details

### Materials

The *Arabidopsis thaliana* genes studied and their corresponding accession numbers: Fim1 – AT4G26700, CHC1 - AT3G11130, CLC2 - AT2G40060, Drp1C – AT1G14830, Tplate- AT3G01780, VHA-a1 - At2g28520, PIN2 - AT5G57090, FLS2 – AT5G46330, Axl2 - AT4G12770, AP2A1 – AT5G22770. All lines used are from *Arabidopsis thaliana*:

35 s::GFP-fABD2 (*Sheahan et al., 2004*), pCLC2::CLC2-GFP and pDRP1C::DRP1C-GFP (*Konopka et al., 2008*); pRPS5A::CLC2-RFP, pLAT52p::TPLATE-GFP x pRPS5A::AP2-RFP and pLAT52::TPLATE-GFP x pRPS5::CLC2-RFP tplate (*Gadeyne et al., 2014*); p35S::Fim1-GFP, p35S:: mTalin-GFP and 35S::ABD2-GFP (*Kost et al., 1998*; *Wang et al., 2004*), p35S::Lifeact-VENUS (lifeact gene was cloned into pK7m34GW to produce a N-terminal fusion of Lifeact to GFP and transformed into Col, a gift from Matyas Fendrych and

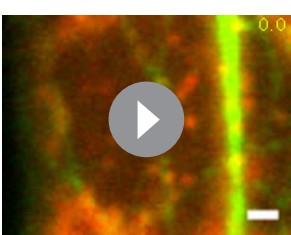

**Video 1.** Actin whipping away the CCVs. Dual channel TIRF-M imaging of hypocotyl epidermal cells expressing ABD2-GFP and CLC2-mKO. Spatial and temporal correlation between appearance of actin subsidiary filaments and the disappearance of CCVs could be seen. Time label is in seconds. Scale bar: 1 μm.

https://elifesciences.org/articles/52067#video1

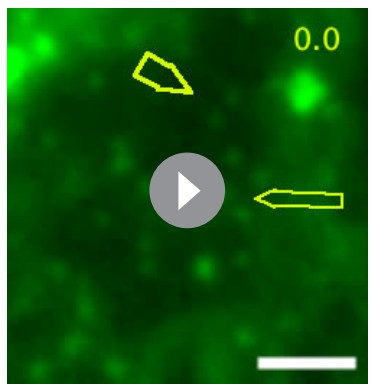

**Video 2.** CCVs losing the coat. TIRF imaging of hypocotyl epidermal cells expressing ST-RFP and CLC2-GFP. CCVs (arrowed) that leave the membrane lose the clathrin coat before reaching EE/TGN. Time label is in seconds. Scale bar: 2 μm.

https://elifesciences.org/articles/52067#video2

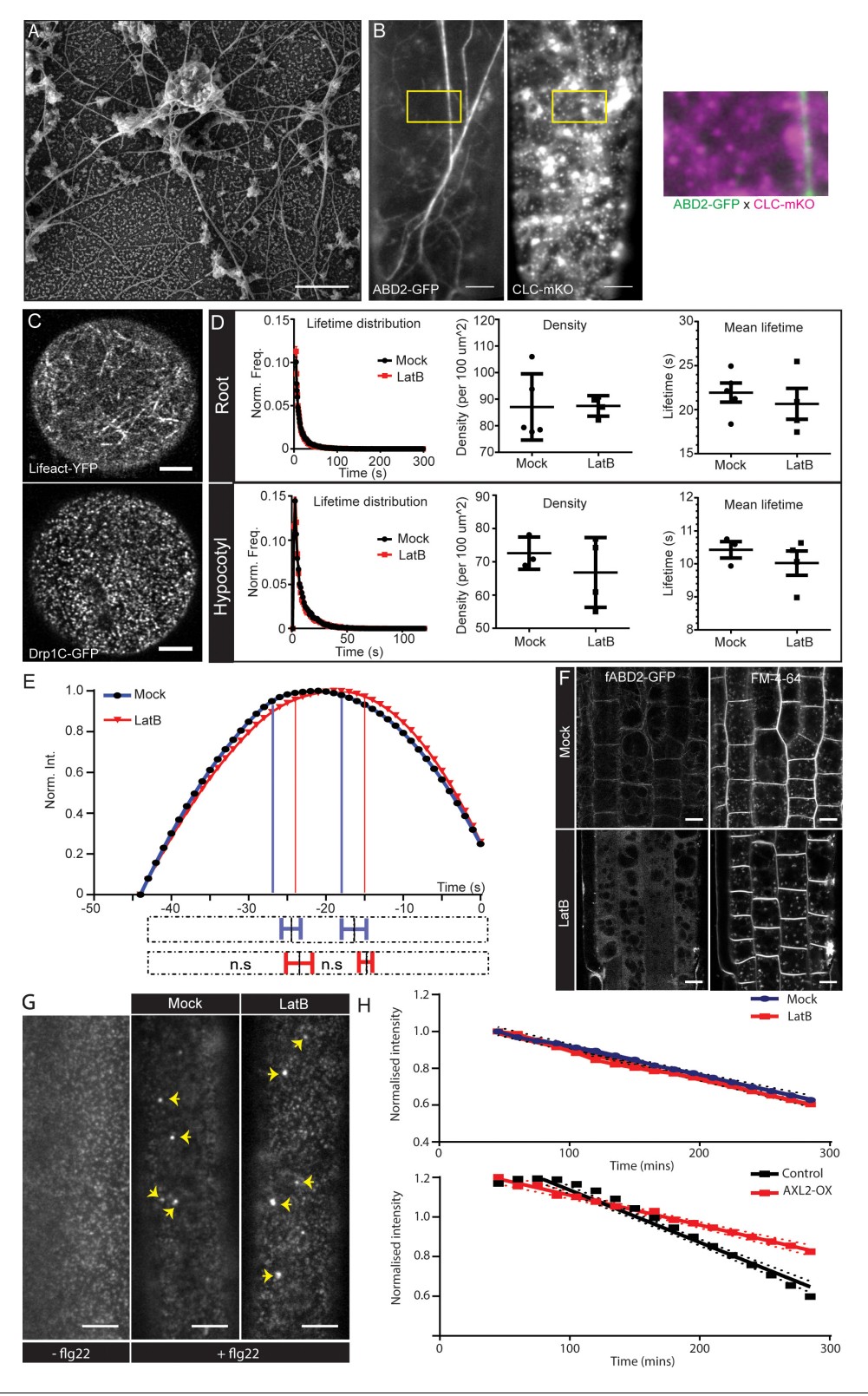

**Figure 3.** Localization and functional importance of actin during endocytosis. (**A**) SEM image of the unroofed protoplasts showing the main and subsidiary filaments of actin in close proximity to the PM. (**B**) Dual channel TIRF image of hypocotyl epidermal cell expressing ABD2-GFP and CLC2-mKO. The insert (yellow box) shows a magnified merge of the channels. Also see Figure S4B. (**C**) Example confocal airy-scan images of root hair cell expressing Lifeact-YFP (left) and the root hair cell expressing DRP1C-GFP (right). (**D**) Normalized lifetime distributions, average lifetime and densities of

*Figure 3 continued*

CLC2-GFP of root (upper panel) and hypocotyl (lower panel) epidermal cells in the absence (black) or presence (red) of LatB (roots; 10 µM, 10 mins, hypocotyl; 10 µM, 1 hr), measured by TIRF-M. Root: mock, 5 cells from individual roots, (32991 tracks), LatB, 4 cells from individual roots, (25517 tracks). Hypocotyl: mock, 3 cells from individual hypocotyls, (10883 tracks), LatB, 4 cells from individual hypocotyls, (12623 tracks). Error bars represent mean ± SEM. Two-sided unpaired T tests found there were no significant differences (Average lifetimes: hypocotyl p=0.43; root p=0.53. Average density: hypocotyl p=0.42; root p=0.95. (See also *Supplementary file 1* table 2 for other markers). (E) Smoothened CCP intensity profile of the mean long-lived CLC2-GFP population in root epidermal cells in the absence (black) or presence (red) of LatB (10 µM, 10 mins). Mock, 6 cells from individual roots, 182 trajectories, LatB, 4 cells from individual roots, 122 trajectories. The extrapolation lines mark the different CCP development phases. The bottom bar plot represents the phase transitions computed individually from the trajectories of each root. The dotted bars represent the whole time course of CCP development; the solid lines with error bars mark the mean ± SD of the transition point between phases. Note that there are no significant differences in the different phases in the presence of LatB. One-sided Mann-Whitney U test; assembly p=0.31, maturation p=0.48 (also see Figure S4D). (F) Confocal microscopy images of root epidermis expressing fABD2-GFP after mock or LatB treatment (20 µM, 30 mins). Actin cytoskeleton is disrupted but FM4-64 is still endocytosed. N = 2 experimental repeats; at least 10 seedlings per condition. (G) TIRF-M images of the hypocotyl epidermal cells expressing FLS2-GFP either with or without flg22 (10 µM, 0.5 hr) treatment, and also cells pretreated with 1 hr of mock or LatB (10 µM, 1 hr) with co-treatment of flg22. Yellow arrows highlight endosomal structures containing FLS2. N = 2 experimental repeats, with nine hypocotyls per condition. (H) PIN2-Dendra endocytic rate, determined by the change of PM PIN2 intensity over time, after LatB (10 µM, 45 mins) compared to mock treatment (top). The dots represent the mean intensity and the dotted lines represent the 95% CI. No significant difference is observed between the slope of the curves; LMER - random effects for position; $\chi^2$- 2.5923; df = 1; p=0.107; N = 2, five seedlings per condition. (Bottom) PIN2-Dendra endocytic rate with the mock induction conditions or induction of AXL2 over-expression (AXL2-OX) for 24 hr (bottom). The slope of the curve for AXL2-OX is significantly lower than control conditions; LMER - random effects for position; $\chi^2$ = 78.095; df = 1; p<2.2e-16 ***; N = 2 experimental repeats, four seedlings per condition; all the epidermal cells in the root meristem were considered. Scale bars; 0.5 µm (A), 5 µm (B,G), 4 µm (C), 10 µm (F).

The online version of this article includes the following video, source data, and figure supplement(s) for figure 3:

**Source data 1.** Source data and code for the quantification for *Figure 3*.
**Figure supplement 1.** Localization and functional characterization of actin during endocytosis.
**Figure 3—video 1.** Dynamics of CCVs after actin perturbation.
https://elifesciences.org/articles/52067#fig3video1
**Figure 3—video 2.** Dynamics of CCVs after actin perturbation.
https://elifesciences.org/articles/52067#fig3video2

---

Moritz Nowack); *p35S::ST-RFP* x *pCLC2::CLC2-GFP* (*Ito et al., 2012*); *pVHA-a1::VHA-a1-GFP* (*Dettmer et al., 2006*); *pFLS2::FLS2-GFP* x *pUBQ10::mRFP-ARA7* (*Beck et al., 2012*); *pPIN2::PIN2-Dendra eir1-1* (*Salanenka et al., 2018*); *XVE >>Axl2* x *pPIN2::PIN2-Dendra* (*Adamowski et al., 2018*); *p35S::ABD2-GFP* x *p35S::CLC2-mKO* was generated by crossing *p35S::CLC2-mKO* (*Naramoto et al., 2010*) with *p35S::ABD2-GFP*; *pPIN2::PIN2-GFP* x *p35S::ARA7-mRFP* was generated by crossing *pPIN2::PIN2-GFP* (*Xu and Scheres, 2005*) with *p35S::ARA7-mRFP* (*Ueda et al., 2004*); *35 s::AP2A1-GFP* x *rPS5A::CLC-tagRFP* was generated by crossing *35 s::AP2A1-GFP* (*Di Rubbo et al., 2013*) with *pRPS5A::CLC-tagRFP* plants and the F1 progeny were used. *pRPS5A:: CHC1-GFP* x *pRPS5A::CLC2-tagRFP* was generated by crossing *pRPS5A::CHC1-GFP* (*Dejonghe et al., 2016*) with *pRPS5A::CLC-tagRFP* plants and the F1 progeny were used.

## Seedling growth conditions

Seeds were sown on ½ MS Agar medium agar, with 1% (w/v) sucrose, stratified for 2–3 days and then grown at 21°C in a 16 hr/8 h day/night cycle. Seedlings to examine roots were grown for 3 or 7 days, seedlings to examine the hypocotyl seedlings were grown for 3 days in dark conditions.

## Root cell culture – maintenance

Arabidopsis (Col-0) root-derived suspension was obtained from Eva Kondorosi (Gif-sur-Yvette, France). The suspension culture is maintained in growth medium (GM) of pH5.7 containing 4.25 g/l of MS salts (Sigma M5524), 30 g/l of sucrose, 0.250 mg of 2,4D, 0.015 mg of kinetin and 2 ml Vitamin B5 stock (100 ml stock contains 0.1 g nicotininc acid, 0.1 g pyridoxine HCl, 1 g thiamine HCl. 10 g myo-inositol).

## Materials used

LatrunculinB, β-estradiol, MS powder and all chemicals used for metal replicas and resin embedding were from Sigma Aldrich; with the exceptions of EM grade glutaraldehyde and pioloform (Agar

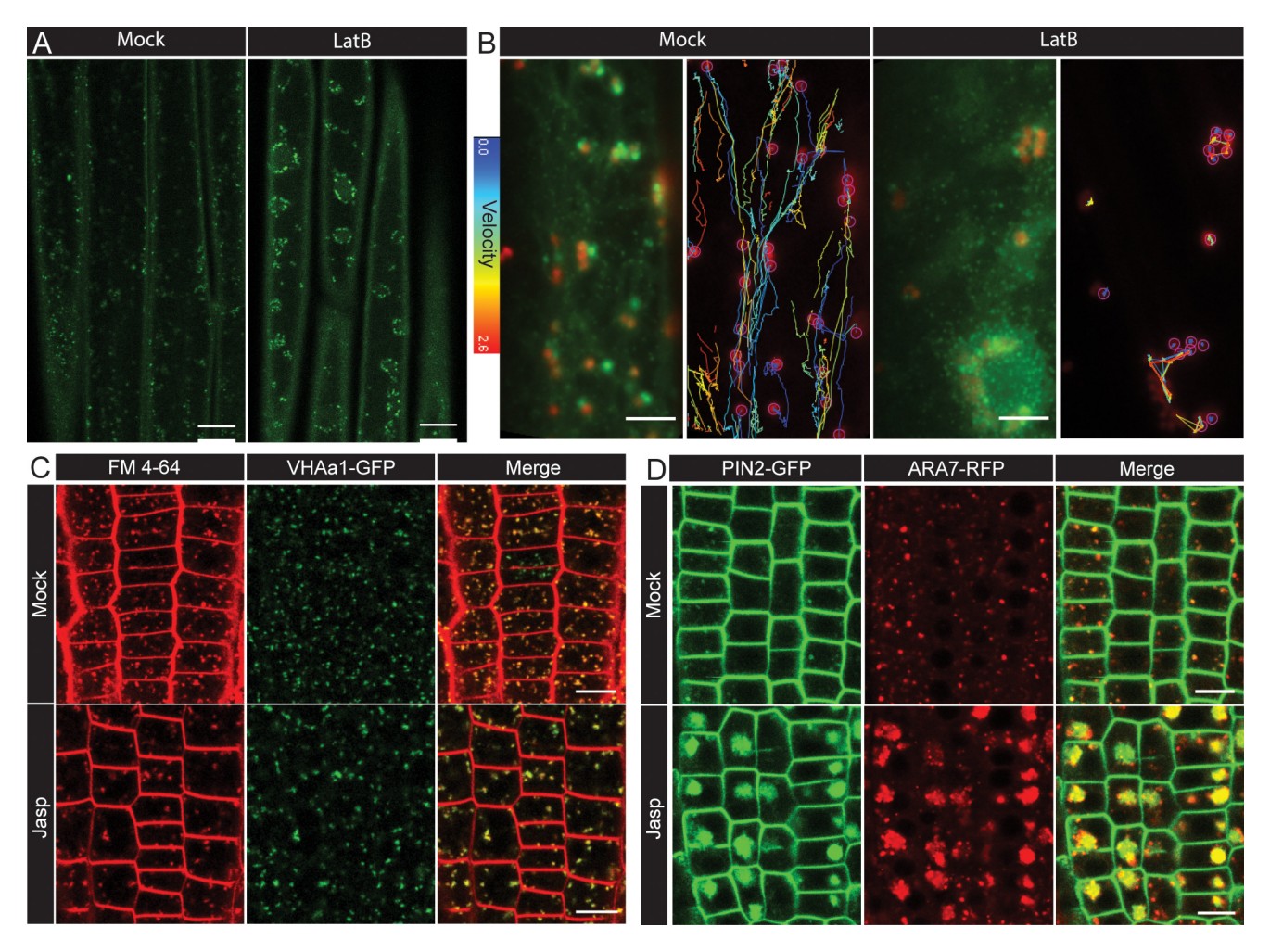

**Figure 4.** Role of actin in post-endocytic trafficking. (**A**) Representative confocal microcopy images of EE/TGN (marked by CLC2-GFP) aggregated after mock or LatB (10 µM, 1 hr) treatment in hypocotyl epidermal cells. Observations were made from five individual hypocotyls per condition. (**B**) Representative TIRF-M images of Golgi (ST-RFP) and EE/TGN (CLC2-GFP) structures (left panel; see also the *Figure 4—video 2*) and the velocity profile by tracking the Golgi (right) after mock and LatB (10 µM; 1 hr) treatment. Observations were made from ≥15 cells per condition. The tracks are very short and belong to low velocity profile after LatB. (**C**) Representative confocal microcopy images of the root epidermis expressing VHA-a1-GFP with either mock or Jasp treatment (5 µM, 30 mins). Observations were make from seven roots per condition. (**D**) Representative confocal microcopy images of the root epidermis expressing PIN2-GFP and ARA7-RFP (marking LE) after mock of Jasp (5 µM, 5.5 hr). Observations were made from eight roots per condition. Scale bars; 15 µm (**A**), 5 µm (**B**), 10 µm (**C, D**).

The online version of this article includes the following video(s) for figure 4:

**Figure 4—video 1.** Actin mediated dynamics of the EE/TGN system.

https://elifesciences.org/articles/52067#fig4video1

**Figure 4—video 2.** Dynamics of the Golgi- EE/TGN after actin perturbation.

https://elifesciences.org/articles/52067#fig4video2

Scientific Ltd., Stansted, UK), Osmium tetroxide (Electron Microscopy Sciences, Hatfield, PA), Acetone and polyethylene glycol (Merck, Darmstadt, Germany), picric acid (Fluka GmbH, Buchs, Switzerland), Uranyl acetate (AL-Labortechnik E.U., Amstetten, Austria). Jasplakinolide was from Santa Cruz Biotechnology, Cellulase from Serva, Macerozyme from Yakult pharmaceuticals, Aclar foil from Ted Pella Inc (Redding, CA), Aluminium planchettes and sapphire disks from Wohlwend (Sennwald, Switzerland), Cryo-vials from Biozym GmbH (Vienna, Austria), Glass cover slips from Roth (Karlsruhe, Germany), Maxtaform H15 finder grids from Science Services GmbH (München, Germany) and Platinum and carbon rods were from Leica Microsystems (Vienna, Austria).

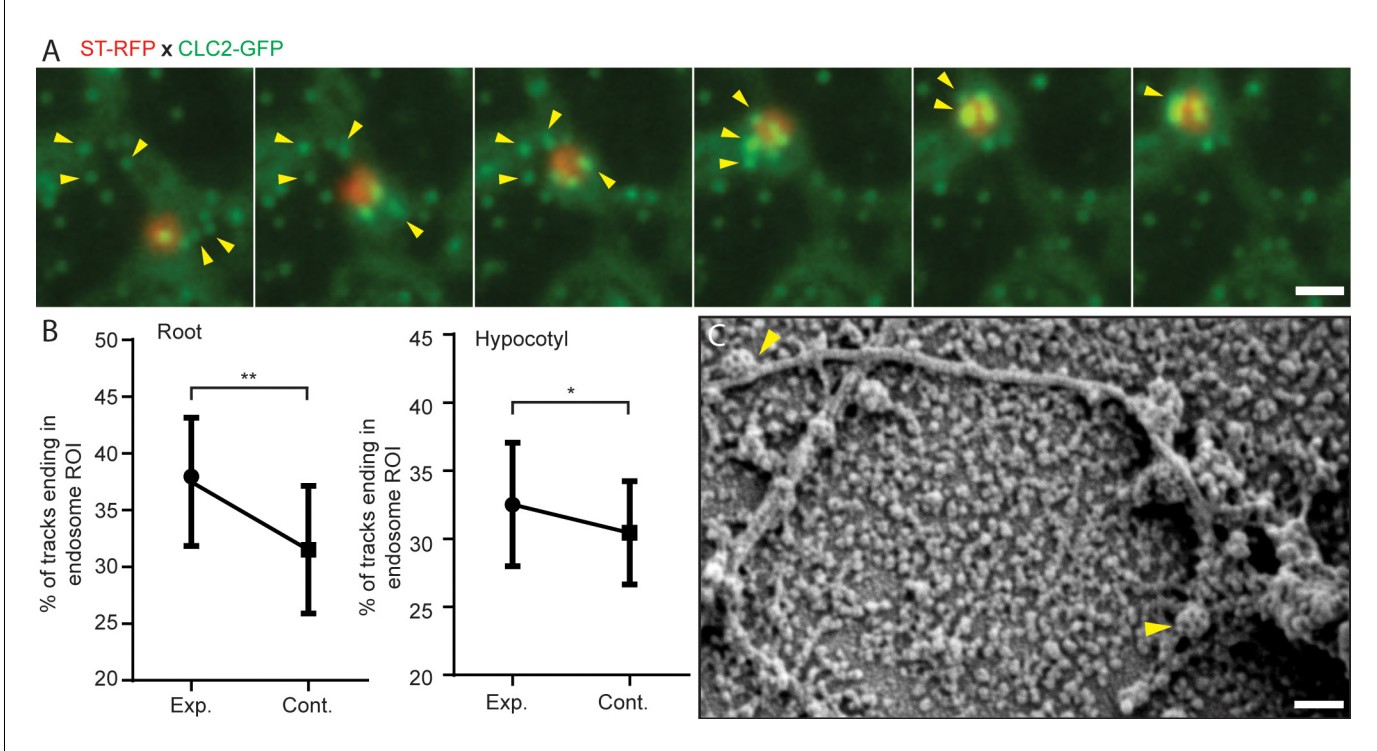

**Figure 5.** Role of actin and early endosome movement in CME. (**A**) TIRF-M images of hypocotyl epidermal cells expressing *ST-RFP* and *CLC2-GFP*. The Golgi apparatus (marked by ST-RFP) which move towards the endocytosed CCVs (marked by CLC2-GFP) are marked by arrows. Time interval: 1.9 s. (See also *Figure 5—video 2*). (**B**) Endosomal pick-up analysis of CCVs positive for both CLC2-tagRFP and TPLATE-GFP in root (left) and hypocotyl (right) cells. Higher proportion of CME events terminates when EE passes by below the PM. N = 6 root cells and 622 endosome tracks, six hypocotyl cells and 1539 endosome tracks. The plot represents the mean ± SD. Paired two-sided t test p=0.0066** for root and p=0.0223* for hypocotyl. (**C**) SEM image of unroofed protoplasts, where arrows depict CCVs attached to a cytoskeleton filament. Scale bars, 1 μm (**A**), 100 nm (**C**).
The online version of this article includes the following video and source data for figure 5:

**Source data 1.** Source data for the quantification in *Figure 5*.
**Figure 5—video 1.** EE/TGN moving along actin picking up an endocytosed CCV.
https://elifesciences.org/articles/52067#fig5video1
**Figure 5—video 2.** Collection of endocytosed CCVs by the EE/TGN.
https://elifesciences.org/articles/52067#fig5video2
**Figure 5—video 3.** Dynamic endosomes enhancing the local rate of CME termination.
https://elifesciences.org/articles/52067#fig5video3

## Methods

### Protoplast preparation and plating

3 day old *Arabidopsis* suspension cultured cells were used. Protoplast isolation from the cells was carried out as described by *Dóczi et al. (2011)* and colleagues. Isolated protoplasts were left at 4°C overnight. They were then plated onto glass coverslips (Ø 12 mm), which were pre-coated with carbon to a thickness of 10 nm and treated with Poly-L-Lysine (PLL) overnight at 4°C and incubated at room temperature for 4 hr. The protoplasts plated onto coverslips were then centrifuged at 800 rpm for 5 mins to aid protoplast adhesion to the coverslips.

### Metal replica electron microscopy

The plated protoplasts were washed in PBS equilibrated to room temperature and the extraction solution (1% Triton X-100 in PIPES-EGTA-magnesium buffer (PEM; 100 mM PIPES, 1 mM EGTA, 1 mM $MgCl_2$, pH 6.9) plus 1% high-MW polyethylene glycol (PEG; 20 kDa) and 2 μM phalloidin ±2 μM taxol) was applied for 4 mins. Samples were then washed three times in PEM plus 1% PEG for 1 min and then fixed in 2% Glutaraldehyde in phosphate buffer (PB) for 20 mins. After washing in distilled

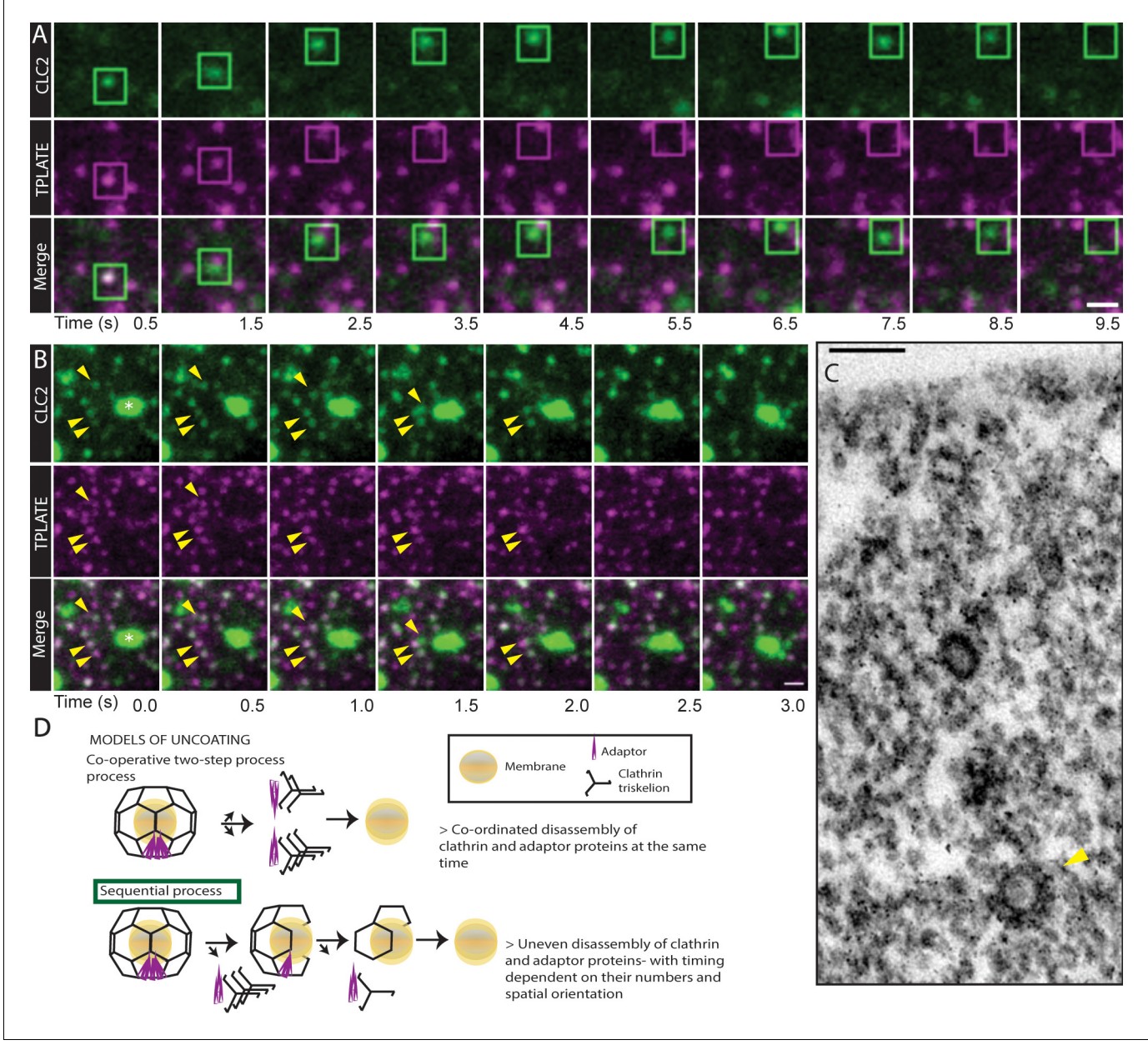

**Figure 6.** Uncoating of CCVs. (**A and B**) TIRF-M images of hypocotyl epidermal cells expressing TPLATE-GFP and CLC2-tagRFP. CLC2 marks both endocytic foci (smaller foci) and the EE/TGN (larger structures). (**A**) The boxed CCV is followed after internalization into cytosol. Time interval: 1 s. See also *Figure 6—video 2*. N = 19 events pooled from 5 cells. (**B**) CCVs indicated by arrows reach the stationary EE/TGN while still containing clathrin (white asterisk). Time interval: 0.5 s. See also *Figure 6—video 1*. (**C**) TEM image of a resin embedded and ultra-thin sectioned protoplast. The yellow arrow denotes a CCV with partially removed coat deeper in the cytosol. (**D**) Models of clathrin coat removal. (top) Co-operative disassembly, where the disassembly of clathrin coat and adaptor TPLATE is temporally coordinated. (bottom) Sequential disassembly, where adaptor is able to disappear entirely before clathrin. Scale bars, 1 μm (**A, B**), 100 nm (**C**).

The online version of this article includes the following video(s) for figure 6:

**Figure 6—video 1.** Uncoating process as the CCVs reach the stationary EE/TGN.
https://elifesciences.org/articles/52067#fig6video1

**Figure 6—video 2.** Uncoating process of a fully developed CCV.
https://elifesciences.org/articles/52067#fig6video2

water, samples were treated with 0.1% tannic acid in water (w/v) for 20 mins at room temperature and 0.2% uranyl acetate in water (w/v) for 20 mins at room temperature. Samples were then 'critical point dried' as described by *Svitkina (2007)* and colleagues. Samples were then fixed on SEM specimen mounts by means of carbon conductive adhesive tabs (Ø 12 mm), and gold-coated to a thickness of 5 nm by rotary shadowing at a 45° angle using a ACE600 coating device (Leica Microsystems).

## Resin embedding ultrastructure analysis

Protoplasts were plated onto Aclar foil, which had been coated with PLL overnight at 4°C, and incubated at room temperature for 4 hr. After a wash in phosphate buffered saline (PBS; 0.1M, 0.9% NaCl, pH 7.4), the protoplasts were incubated in one of the following solutions for 30 mins at room temperature: GM buffer or hyperosmolar buffer (75 mM Mannitol in GM buffer). Protoplasts were then fixed in 2% formaldehyde plus 2.5% glutaraldehyde and 15% of a saturated solution of picric acid in phosphate buffer (PB; 0.1M, pH 7.4) for 30 mins at room temperature. The protoplasts were then washed with PB buffer and then incubated with 0.5% tannic acid in PB (w/v) for 1 hr at 4°C, then 1% osmium tetroxide in PB (w/v) for 30 mins at 4°C and then 1% uranyl-acetate in water (w/v) overnight at 4°C. Protoplasts were then contrast-enhanced with Walton's lead aspartate for 30 mins at 60°C, dehydrated in graded ethanols and anhydrous acetone, and embedded in epoxy resin (Durcupan ACM). Serial ultrathin sections of 40 nm were cut with an ultramicrotome UC7 (Leica Microsystems), collected onto Formvar-coated copper slot grids and stained with 1% aqueous uranyl acetate and 0.3% lead citrate.

## Treatment conditions

All the treatments were carried out at room temperature in ½ MS medium containing 1% (w/v) sucrose. Jasp and LatB treatments of the seedlings were carried out by dissolving the compounds in DMSO, and then diluting to the working concentration with liquid 1/2 MS and 1% sucrose solution. Specific concentrations and duration of treatments are defined in the relevant figure legends. Throughout imaging experiments, the seedlings were kept in the mock or treatment solutions, with the exception of FM4-64 uptake experiments. To induce the overexpression of Auxilin2, 2 day old seedlings expressing *XVE >>Axl2 x pPIN2::PIN2-Dendra* were induced by transferring them to solid ½ AM with 1% sucrose plate supplemented with 2 µM β-estradiol. The seedlings were continuously under chemical induction during subsequent imaging.

To examine the uptake of FM4-64, following the actin/mock treatment, 3 day old seedlings were incubated with 2 µM FM4-64 dissolved in liquid ½ MS with 1% sucrose medium for 2 mins. The seedlings were then washed twice before being imaged. To examine Flagellin uptake, 3-day-old etiolated seedlings were incubated in 10 µM flg22 in liquid ½ MS with 1% sucrose medium for 30 mins. Flagellin treatment was continuously present during imaging.

## Microscopy

### Confocal microscopy

To determine the endocytic rate of *PIN2*, photo conversion and imaging of photo converted PIN2-Dendra at the PM was conducted using a Zeiss LSM700 vertical confocal microscope equipped with a Plan-Apochromat 20x/NA 0.8 air objective; as previously described (*von Wangenheim et al., 2017*). The whole root was photo converted from green to red by a 2 mins excitation by UV light, from a mercury arc lamp and through a DAPI filter. The growing root was tracked for 4 hr with the 'Tip Tracker' software (*von Wangenheim et al., 2017*) and the loss of intensity of the converted red PIN2-Dendra signal was imaged through all roots at 15 min intervals over different planes of the entire epidermal tissue layer. Imaging of FM4-64 dye uptake, PIN2 co-localization experiments in root meristem, and imaging of CLC2-GFP localization in etiolated hypocotyls were done using a LSM700 inverted confocal microscope; with a Plan-Apochromat 40x/NA 1.3 water objective. Root hairs were imaged with a Zeiss LSM880 upright confocal system, using a Plan-Apochromat 40x/NA 1.2 water objective and AiryScan detector.

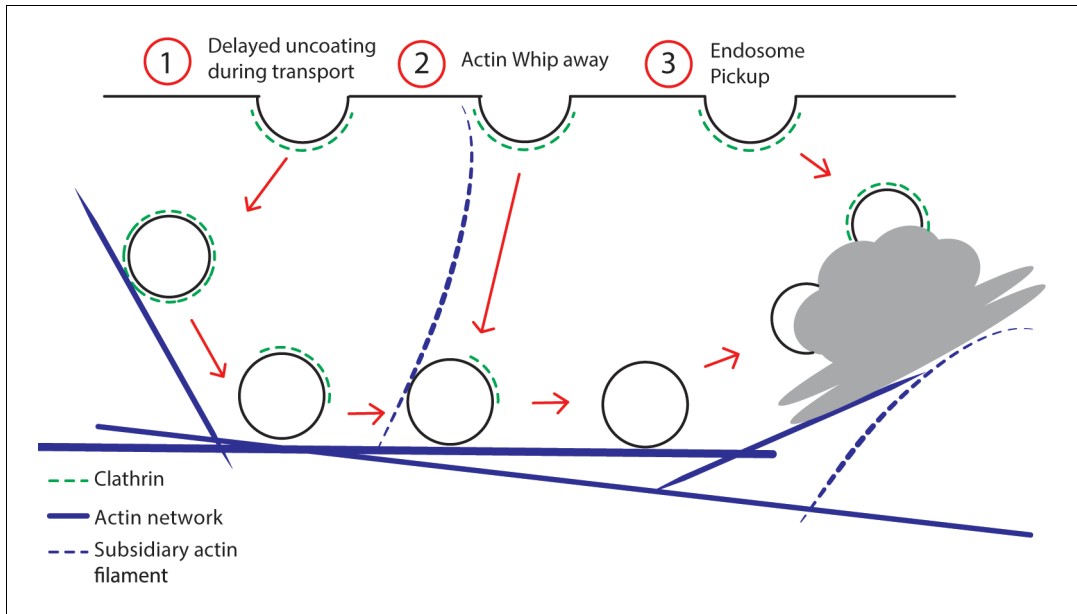

**Figure 7.** Model for post-endocytic trafficking of CCVs. Trafficking of the fully developed CCVs and the EE/TGN compartments along actin represented in different scenarios 1) Delayed sequential uncoating of the CCV during the transport to the EE/TGN 2) Actin subsidiary filament whipping away the CCV after scission transporting it to the EE/TGN 3) EE/TGN directly picking up the CCV after scission.

## TIRF microscopy

Roots of 7-day-old seedlings or hypocotyls of 3-day-old etiolated seedlings were imaged with a Olympus IX83 inverted microscope equipped with a CellTIRF module using a OLYMPUS Uapo N 100x/1.49 Oil TIRF objective. A quad line beam splitter emission filter (Chroma) in combination with an EM-CCD camera (Hamamatsu) was used to collect images. Excitation wavelengths were 488 for GFP tags and 561 nm for RFP tags. Time-lapse imaging in the hypocotyl was done on the epidermal cells that are closet to the root-hypocotyl junction, after cutting the cotyledon. Single channel images were collected at two frames per second. Dual channel imaging was done sequentially at the frame rate as specified in the figures/movies.

Roots were prepared for imaging as previously described by *Johnson and Vert (2017)* with an additional step of sealing the coverslip on to a microscope slide with nail polish. Single channel images were collected at one frame per second for either 301 or 601 frames. Dual channel images were collected sequentially in the GFP and RFP channels at a rate of 1 frame per second for 601 frames.

## Electron microscopy

To examine the metal replicas of unroofed protoplasts a Zeiss FE-SEM Merlin VP Compact was used. An accelerating voltage of 0.5–5 kV was applied under high vacuum condition and In-lens duo detector was used for image recording. For transmission electron microscopy of resin embedded samples, the protoplasts were examined and imaged at an accelerating voltage of 80kV in a Thermo Ficher TEM-Tecnai10 equipped with EMSIS Megaview G3 camera.

## Processing and quantification

### CCSs population ultrastructural analysis

The classification of the SEM images of the CCSs after metal replica was done manually using Fiji (NIH) and plotted using R. For all the *en face* surface-view analysis, only the CCSs with the entire view of the basket were considered; and the baskets that were partially visible or lying on the side

were dismissed. To study the invagination of CCSs at the PM, TEM images of CCSs after resin embedding and sectioning of protoplasts were visually examined and classified using Fiji (NIH).

## TIRF-M image analysis

Time series movies were subjected to just the particle detection and tracking programs of the cmeAnalysis package, as previously described (*Johnson and Vert, 2017*). The frame rate of data acquisition determines the temporal certainly of the mean lifetimes reported. For single channel analysis; only tracks which were not present in the first or last 10 frames, or within 10 pixels of the edge of the movie and persisted for 5 of more frames were considered for further analysis. For dual channel analysis, a master/slave detection approach was used, where the master had to satisfy the aforementioned criteria, but also required the slave channel to be present for more than five frames. Density calculations were made by imposing a 100 × 100 pixel region of interest (ROI) in the center of the movie. All the tracks within this ROI were used to produce an average density over 100 frames of the movie, where the middle frame was used as a reference and 50 frames before and after were used. To produce fluorescence profiles of the movies, all tracks from the experiments were combined, and tracks with the overall mean + /- 3 frames were selected, and their fluorescence was normalized and combined to produce a mean fluorescence profile. For dual channel images, to produce a departure plot, the master tracks mean + /- 3 frames were aligned to their end point, and the slave channel fluorescence signal was also plotted. The endosome pickup analysis was conducted by first tracking the endocytosis spots with the single channel TIRF-M image analysis, and then tracking the larger endosome spots with TrackMate (*Tinevez et al., 2017*). To accurately track the endosomes, some of the movies were subjected to histogram matching bleaching correction. Only endosomes, which had a trajectory displacement greater than 10 pixels and not present within the first and last 10 frames of the movie were used for analysis. A 30 pixel ROI was created which centered on the co-ordinates of the endosome, and followed the endosome trajectory over time. Endocytosis tracks present in this mobile ROI were counted and the percentage of tracks ending in this ROI were calculated. To generate control endosome track; endosomes were labeled with a identifying number based on their order of detection, if the endosomes first co-ordinate was located in the upper left quarter of the image, 50 pixels were add to its x axis if its identifying label was odd, or 50 pixels to its y axis if it was an even labeled track. If the endosome was located in the upper right quarter of the movie, 50 pixels were subtracted from its x co-ordinates if it was an odd track, even tracks had 50 pixels added to its y-axis. Tracks located in the bottom quarters of the movie had these additions of subtractions to the endosome coordinates reserved.

For co-localization, 10 frames of the detections made by the cmeAnalysis package were used to make a median z projection. Then the comDet plugin (https://imagej.net/Spots_colocalization_(ComDet)) for Fiji was used to calculate a co-localization value.

## Quantification of intensity profile

The intensity time-course of the endocytic foci marked by CLC2-GFP was processed as described in *Loerke et al. (2009)*, where the duration of the developmental phases were characterized based on the average change of intensity of the average trajectories over time. The average intensity values of CLC2-GFP measured over time in each root were normalized and subjected to smoothening, using a lowess curve. The phase transition was determined at the point when the slope drops 20% below the minimum slope. Post processing were made in Graphpad PRISM6.

## Population fitting

From the lifetimes of CLC2 tracks obtained, we first generated a normalized density histogram, $h(t)$, where $t$ denotes different possible lifetimes. As the only tracks with a lifetime of at least five frames were considered for analysis, the lifetimes have a specific lower bound $t_{low}$ = 5*frame rate. The lifetimes also have an upper bound $t_{high}$, corresponding to the total imaging time. Hence, we have $h(t)$ =0, for $t < t_{low}$ and $t > t_{high}$. We choose a specific bin size, $\Delta t$, equal to the inverse of the frame rate, and obtain $h(t)$ as a discrete density function over a set of $N$ lifetimes (Adamowski and Friml), $1 \leq i \leq N$, uniformly separated by $\Delta t$, between $t_1 = t_{low}$ and $t_N = t_{high}$. Hence, $h(t_i)\Delta t$ denotes the probability

that the lifetime is in the interval [$t_i, t_{i+1}$). We normalized the histogram to ensure the following normalization condition:

$$\sum_{i=1}^{N} h(t_i)\Delta t = 1.$$

As this distribution appears to arise from a mixture of exponential distributions. Therefore, to further characterize the distribution and extract various subpopulations that have different biological relevance, we fitted this distribution to a mixture of $n$ (truncated) exponential distributions. Each exponential subpopulation has a rate and fractional contribution to the mixture as free parameters. Hence, the free parameters to be inferred in the overall fitting function are the rates, $\lambda_i$, and the fractional contributions, $\mu_i$, $i \leq n$. As the sum of fractional contributions sum to 1, $\sum \mu_i = 1$, only $n$-1 fractional contributions need to be inferred. For a fitting of $n$ exponential distribution mixture, the total free parameters to be inferred are $k_{exp}(i)=2$ $n$-1. Each exponential distribution is truncated to be bounded between $t_{low}$ and $t_{high}$, resulting in the following truncated density function for the $i^{th}$ subpopulation:

$$f_i(t) = \frac{pdf_i(t)}{cdf_i(t_{high}) - cdf_i(t_{low})} = \frac{\lambda_i e^{-\lambda_i t}}{e^{-\lambda_i t_{low}} - e^{-\lambda_i t_{high}}},$$

where $pdf_i(t)$ and $cdf_i(t)$ correspond to the probability density function and the cumulative distribution function of the $i^{th}$ exponential subpopulation. The overall fitting function takes the following form:

$$\hat{h}(t) = \sum_{i=1}^{n} \mu_i f_i(t),$$

with the additional constraint that $\sum \mu_i = 1$. To perform the fitting, we minimize the following cost function:

$$\Lambda(\{\lambda_i, \mu_i\}) = \sum_{i=1}^{N} \left(h(t_i) - \hat{h}(t_i)\right)^2,$$

over the various possible parameter values $\lambda_i > 0$, and $0 \leq \mu_i \leq 1$. This approach fits the normalized histogram density of the dataset to a mixture of truncated exponential density functions. To make sure that this constrained minimization problem does not result in local minima and return suboptimal parameter sets, we perform the minimization from various initial conditions and pick the best fit among the whole set of minimizations. In summary, we obtain

$$\{\lambda_i^*, \mu_i^*\} = \underset{\lambda_i, \mu_i}{\operatorname{argmin}} \Lambda(\{\lambda_i, \mu_i\}),$$

with the value of the cost function at the inferred minimum being $\Lambda^* = \Lambda(\{\lambda_i^*, \mu_i^*\})$. The values of AIC and BIC are obtained as $AIC = 2k_{exp} + N \ln(\Lambda^*/N)$ and $BIC = k_{exp} \ln N + N \ln(\Lambda^*/N)$. From the inferred fractions of the subpopulations, $\mu_i$, one can obtain the "true" contributions by correcting for the skewing that results from the truncation in the lifetimes to $t_{low}$ and $t_{high}$. Let the true fraction of the $i^{th}$ subpopulation be $\mu_i^0$. Also, define $\omega_i = cdf_i(t_{high}) - cdf_i(t_{low})$. We have the following for every $i$:

$$\mu_i = \frac{\omega_i \mu_i^0}{\sum_{j=1}^{n} \omega_j \mu_j^0},$$

from which we can obtain the true fractions $\mu_i^0$ by solving a system of linear equations. We also performed fitting with a mixture of exponential and Rayleigh distributions instead of just exponential distributions, by making sure that the $i^{th}$ subpopulation is defined by a $pdf$ and $cdf$ that correspond to the Rayleigh distribution. The clathrin lifetime histogram subjected to Rayleigh and exponential distribution produced fittings for up to 5 populations. A previous study (*Loerke et al., 2009*) in mammals used Weibull distributions to fit clathrin pit lifetime data. However, fitting of a simulated

dataset indicated that Weibull distributions overfit lifetime data, and do not infer the true underlying parameters corresponding to the subpopulations.

### PIN2 endocytic rate test

The time series images were processed using FIJI (NIH). Maximum intensity Z-projections of the epidermal PIN2 signal were used for analysis. A ROI covering majority of the meristem was drawn and the mean intensity of the photo-converted PIN2 signal was measured over time.

### Velocity profile by organelle tracking

The time-lapse movies made on the movement of Golgi apparatus were processed using the Fiji plugin TrackMate (*Tinevez et al., 2017*). Tracks were profiled according to the maximum velocity exhibited by the organelle.

### Statistical analysis

Statistical analysis on data for CCP developmental progression, endocytic density and lifetimes was conducted using Graphpad Prism 6. Significance is defined by $p < 0.05$.

Statistical analyses for PIN2 internalization rate between treatments were carried out using R version 1.1.383. We used a linear mixed effects regression (LMER) to test for the effect of treatment on PIN2 internalization rate. We modeled PIN2 PM intensity values as a function of two predictors: time and treatment and their interaction, and we included a random intercept for each root, which is common for longitudinal studies (*Bolker et al., 2009*). We assessed the model significance comparing it to a null (mean) model, and the significance of the interaction comparing to a model without interaction, using likelihood ratio tests. The modeling package lme4 was used (*Bates et al., 2015*). The model assumptions were checked by 1) testing for equal variance of the residuals 2) testing for normality of the residuals and 3) testing the normality of the random effects.

The number of samples and the repetitions of each experiment are all described in the respective figure main text and legends.

## Acknowledgements

We thank Sebastian Bednarek, Daniel van Damme, Matyas Fendrych, Moritz Nowack, Hongjiang Li, Markus Geisler, and Jenny Russinova for gifting us material; Maciek Adamowski for valuable discussions; the life science facilities at IST Austria, especially the entire team of the entire team of IST electron microscopy facility for their technical support; Lenka Matejovicova for her help with statistics. We also thank, the IST Austria Bioimaging facility for the assistance with the microscopes. This project has received funding from the European Research Council (ERC) under the European Union's Horizon 2020 research and innovation programme (grant agreement No 742985) and Austrian Science Fund (FWF): I 3630-B25. We would like to dedicate this publication to the memory of Chris Hawes, whose enthusiasm about plant cell biology and his work was a great inspiration to this work.

## Additional information

### Funding

| Funder | Grant reference number | Author |
|---|---|---|
| H2020 European Research Council | 742985 | Madhumitha Narasimhan Jiří Friml |
| Austrian Science Fund | I3630B25 | Alexander Johnson Jiří Friml |
| European Molecular Biology Organization | ALTF 723-2015 | Shutang Tan |

The funders had no role in study design, data collection and interpretation, or the decision to submit the work for publication.

## Author contributions
Madhumitha Narasimhan, Alexander Johnson, Conceptualization, Formal analysis, Investigation, Methodology; Roshan Prizak, Barbara Casillas-Pérez, Formal analysis; Walter Anton Kaufmann, Methodology; Shutang Tan, Investigation; Jiří Friml, Conceptualization, Supervision, Funding acquisition, Project administration

## Author ORCIDs
Madhumitha Narasimhan https://orcid.org/0000-0002-8600-0671
Alexander Johnson https://orcid.org/0000-0002-2739-8843
Shutang Tan http://orcid.org/0000-0002-0471-8285
Jiří Friml https://orcid.org/0000-0002-8302-7596

## Decision letter and Author response
Decision letter https://doi.org/10.7554/eLife.52067.sa1
Author response https://doi.org/10.7554/eLife.52067.sa2

# Additional files

## Supplementary files
- Supplementary file 1. Supplementary tables.
- Transparent reporting form

## Data availability
Source data files have been provided for Figures 1, 2, 3, and 5.

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
