## [Decision Letter]

Thank you for submitting your article "Evolutionarily unique mechanistic framework of clathrin-mediated endocytosis in plants" for consideration by *eLife*. Your article has been reviewed by three peer reviewers, and the evaluation has been overseen by a Reviewing Editor and Suzanne Pfeffer as the Senior Editor. The following individual involved in review of your submission has agreed to reveal their identity: Fernando Aniento (Reviewer #2).

The reviewers have discussed the reviews with one another and the Reviewing Editor has drafted this decision to help you prepare a revised submission.

There is a general consensus regarding not only the substantial and broad interest of your results, but also the thoroughness of the work. In particular, all reviewers agreed on the importance of the experiments, showing that actin is dispensable for endocytic uptake but rather plays a role in post-internalization events. Other aspects of the work showing the evolutionary divergence of the mechanism of clathrin-mediated endocytosis in plants, as compared to mammals or yeast, were also very much appreciated. Nevertheless, there are some concerns regarding the analysis of the electron microscopy images and the conclusions derived, which you should take into consideration for revision.

Essential revisions:

1) The characterization of the "hexagon" versus "pentagon" basket is puzzling. All coats will have 12 pentagons (a mathematical rule for a closed polygon). Size is increased by adding hexagons. It does not make sense to talk about a pentagon basket, as they are all pentagon baskets. If the authors are characterizing baskets with different numbers of hexagons, that would be a better way to describe their data. Given that the two types of baskets identified by the authors are the same size, could these not be the same basket from different angles? If not, better to categorize by numbers of hexagons or even by numbers of clathrin molecules in the basket.

2) The authors make a very strong conclusion about the plant cells using the constant curvature model to assemble their endocytic vesicles. However, the arguments about constant curvature versus re-arrangement need to be supported by measurements of the theta angle and TEM, as was done in Avinoam et al., 2015. Looking at Figure 1—figure supplement 1D, the theta angle looks like it's changing, which is not expected in the constant curvature model. Looking at pentagons and hexagons does not provide this information (see point 1). The authors should also be aware that recent studies suggest intermediate models, where the coat area keeps growing during vesicle budding, but the coat curvature still changes (Bucher et al., 2018; Scott et al., 2018).

3) The lack of large clathrin plaques in EM is interesting. However, the fluorescence studies show some apparently large areas of clathrin accumulation (for example in Figure 2). Can the authors somehow verify that those large structures are not flat lattices?

4) The EM is done in protoplasts, whereas the live cell imaging is done in intact cells. These are quite different growth condition and endocytosis could also behave differently, which would complicate the interpretation of the data. Could the authors do live cell imaging in protoplasts or EM in tissues?

5) Providing background on Arabidopsis clathrin would be helpful to understand whether the dynamic labeling using the clathrin light chain 2 (CLC2) is representative of all clathrin or just a subset of coated pits. There is the concern, particularly for uncoating, that measuring clathrin dynamics with a single tagged subunit could influence this process. Could an additional clathrin marker (eg CHC) be used to support the results? If the GFP tagged CHC does not complement, the authors should at least present co-staining of the CHC and the GFP-tagged CLC2.

6) The authors need to demonstrate that the used latB concentration lead to depolymerisation of the actin structures in their experimental system.

[Editors' note: further revisions were suggested prior to acceptance, as described below.]

As previously indicated, there is consensus related to the great interest of your results and the thoroughness of this study. In particular, all reviewers agreed on the importance of the experiments showing that actin is dispensable for endocytic uptake but rather plays a role in post-internalization events, as well as on the evolutionary divergence regarding the endocytic protein dynamics and the mechanism of clathrin assembly and un-coating in plants, as compared to mammalian cells or yeast. The reviewers agree now that the manuscript reads better than the first version and some important issues have been addressed. However, they still have some concerns that you need to address in the text:

You still classify the clathrin coated structures to "hexagonal basket-type" and "pentagonal basket-type". It remains unclear why you think this is a useful classification. All clathrin coated vesicles are expected to have both hexagons and pentagons, and therefore, observing one or the other on the vesicle surface would not be informative. You need to indicate how the lattice is built in the text (always 12 pentagons with different numbers of hexagons). In keeping with this, your data indicate that the "pentagonal pits" are smaller in diameter, likely because they have fewer hexagons. This needs to be clarified also.

The data showing that CLC2 labels 60% of CHC1 coated pits is a good addition, but now, acknowledging that there are two types of CHC's further complicates interpretation of the data. It would be more accurate if you can clearly state that the live cell imaging data can technically be applicable to only one type of clathrin. Or maybe CHC1 represents most of the clathrin in these cells? Further clarification of whether CLC2 can associate with both CHCs or just one should be explicit in the text so it is apparent how much of the clathrin in plant cells is represented in the live cell imaging.

The issue of different CHC's doesn't change the conclusions about the actin association with pits, but it does change the interpretation of the uncoating results. Looking at CLC2 vs adaptors means that you could be looking at more than one type of clathrin (CHC1 or CHC2 with CLC2 bound to either CHC). Maybe CHC1 clathrin dissociates faster than CHC2 clathrin, so the residual CLC2 represents a different clathrin uncoating by a different mechanism. Again, the fact that the uncoating dynamics in plants appear different to yeast and mammals is still valid, but the possibilities of more than one type of clathrin with potentially different behaviors needs to be acknowledged and discussed in full to ensure that the reader understands the implications.

---

## [Author Response]

Essential revisions:1) The characterization of the "hexagon" versus "pentagon" basket is puzzling. All coats will have 12 pentagons (a mathematical rule for a closed polygon). Size is increased by adding hexagons. It does not make sense to talk about a pentagon basket, as they are all pentagon baskets. If the authors are characterizing baskets with different numbers of hexagons, that would be a better way to describe their data. Given that the two types of baskets identified by the authors are the same size, could these not be the same basket from different angles? If not, better to categorize by numbers of hexagons or even by numbers of clathrin molecules in the basket.

We thank the reviewers for highlighting the potential for confusion caused by the terminology we have used to characterize the whole clathrin basket. We have altered the manuscript to make our classifications clearer to the readers. We have re-written this section and the added a graphical illusion in Figure 1. Briefly, the whole clathrin basket is characterized into ‘pentagonal’, ‘hexagonal’ or ‘irregular’ baskets based on the surface-view, where the number of rings around a central ring is used to make this determination. For example, if 5 rings are counted around a central ring, it is a pentagonal CCP appearing to form a pentagon shaped.

During this quantification, care was taken to exclude any CCS which appeared to be lying on its side or appearing at an angle. But to alleviate the reviewer’s concerns that the basket shape (hexagonal/pentagonal) recorded could be due to the viewing angle, we have included details about this point in the Materials and methods section of the text to highlight this concern.

The reason we have not quantified the individual clathrin hexagonal/pentagonal rings or the number of clathrin molecules themselves is that they cannot clearly be identified in our SEM replica images. Therefore, we chose to look at the size and shape of the entire basket as this was a much more reliable component to accurately measure.

2) The authors make a very strong conclusion about the plant cells using the constant curvature model to assemble their endocytic vesicles. However, the arguments about constant curvature versus re-arrangement need to be supported by measurements of the theta angle and TEM, as was done in Avinoam et al., 2015. Looking at Figure 1—figure supplement 1D, the theta angle looks like it's changing, which is not expected in the constant curvature model. Looking at pentagons and hexagons does not provide this information (see point 1). The authors should also be aware that recent studies suggest intermediate models, where the coat area keeps growing during vesicle budding, but the coat curvature still changes (Bucher et al., 2018; Scott et al., 2018).

We agree with the reviewers that approach used by Avinoam et al., is a good way to determine the ‘radius of curvature’ during CCP development. However, during current existing plant TEM protocols, the preservation of invaginations on the PM is incredibly rare. Over decades of TEM work on endocytosis in plants, there is only limited number to events recorded and this also applies to our own TEM efforts. Therefore, it is impossible to make a real quantitative analysis such as ‘the radius of curvature’, and any conclusion from our very few TEM section images are hard to justify.

To overcome this issue, we therefore have to rely on a different approach to address the question of membrane re-modeling during CCV formation using the replica protocol; as this is the only EM method in plant cells which produces intact CCSs with any reliability and quantity. Therefore, we relied on the characterization of the whole basket view (not of individual polygons, as detailed in response one, and now clarified in the main text). Using this surface-view approach, we can make an approximation of the ‘radius of curvature’ of all the CCPs, which we found to be predominantly constant, thus pointing to constant curvature model. We have explained our methodology more clearly in the text and Figure 1 and Figure 1—figure supplement 1.

Avinoam et al., measured theta, the angle between PM and the invaginating vesicle, as a proxy to stages of CCP development which changes the same way for both the models. To try and analyze our CCSs in a similar fashion to the theta angle approach demonstrated by Avinoam et al., we assume theta to represent the degree of invagination. Therefore, by assessing the side-view of each CCP, we were able to determine an approximation of the degree of invagination. Using both the side-view analysis of degree of invagination, and the surface-view analysis of the baskets, we have estimated the ‘radius of curvature’ of different stages of CCPs. We have presented the quantifications in Table 1 in Supplementary file 1 and explanations in the main text.

In the intermediate models highlighted by the reviewer, clathrin is required to first assemble into flat lattices on the PM and then invaginate inwards, with the addition of more clathrin. Because of this requirement, we loosely classified the constant area and intermediate models together (Bucher et al., 2018). In our analysis, there were <4% CCSs that were flat, which could potentially develop and form CCVs following either ‘intermediate’ or ‘constant area’ model (as the reviewer pointed out); which we have now discuss in the text.

3) The lack of large clathrin plaques in EM is interesting. However, the fluorescence studies show some apparently large areas of clathrin accumulation (for example in Figure 2). Can the authors somehow verify that those large structures are not flat lattices?

The reviewers are correct to point out that there are large clathrin structures in the live plant samples. However, they do not represent static lattices on the PM as when looking at the dynamic movies of the live plant samples (Figure 4B, Videos 4, 7 and 8), these are highly mobile elements bellow the PM. They are mobile EE/TGN structures and are not PM associated, like those seen in the EM work of Figure 1F.

To clarify this, in the text we have referred to the movies showing the motility of these structures in the main text. For example, in Figure 2 we have referred the supplemental videos.

4) The EM is done in protoplasts, whereas the live cell imaging is done in intact cells. These are quite different growth condition and endocytosis could also behave differently, which would complicate the interpretation of the data. Could the authors do live cell imaging in protoplasts or EM in tissues?

We agree that there indeed could be difference in endocytic processes in protoplasts and intact plant tissues. Unfortunately, the metal replica approach used here has been developed and optimized for the use of single cells and remains a technical challenge to successfully work with intact plant tissues. While TIRF-M has been used on some types of plant protoplasts (for example, BY2 cells), the *Arabidopsis* root culture cells (as used in the replica EM experiments) are much smaller and rounder. When we tried to image them, we were unable to successful establish a protocol which allowed reproducible TIRF imaging and analysis to be conducted, which was most likely due to their small surface area on the coverslip and the roundness of their overall shape.

While there may be potential differences in the vesicle formation in protoplast and plants, it is important to note that the size of the CCVs quantified in our EM work closely matches the size of CCVs purified from whole plant tissue extracts (Reynolds et al., 2014 and Mosesso et al., 2019), thus suggesting that the same mechanisms are functional in both tissues and single cells. We have now included this sentiment in the main text.

5) Providing background on Arabidopsis clathrin would be helpful to understand whether the dynamic labeling using the clathrin light chain 2 (CLC2) is representative of all clathrin or just a subset of coated pits. There is the concern, particularly for uncoating, that measuring clathrin dynamics with a single tagged subunit could influence this process. Could an additional clathrin marker (eg CHC) be used to support the results? If the GFP tagged CHC does not complement, the authors should at least present co-staining of the CHC and the GFP-tagged CLC2.

We have included some more information about *Arabidopsis* clathrin in the Introduction of the paper. We also included new data. Using plants co-expressing CLC2-FP (1 of the 3 *Arabidopsis* clathrin light chain isoforms) and CHC1-FP (1 of the 2 clathrin heavy chain isoforms), we have determined the co-localization to be around 60% and included this new data in the text and as a Figure 2—figure supplement 1.

6) The authors need to demonstrate that the used latB concentration lead to depolymerisation of the actin structures in their experimental system.

We agree that this is a critical point, and as such, we have highlighted the depolymerisation effect for both roots and hypocotyl after LatB in Figure 3C and Video 2. We show that the concentrations used were effective in the experimental systems.

[Editors' note: further revisions were suggested prior to acceptance, as described below.]

As previously indicated, there is consensus related to the great interest of your results and the thoroughness of this study. In particular, all reviewers agreed on the importance of the experiments showing that actin is dispensable for endocytic uptake but rather plays a role in post-internalization events, as well as on the evolutionary divergence regarding the endocytic protein dynamics and the mechanism of clathrin assembly and un-coating in plants, as compared to mammalian cells or yeast. The reviewers agree now that the manuscript reads better than the first version and some important issues have been addressed. However, they still have some concerns that you need to address in the text:You still classify the clathrin coated structures to "hexagonal basket-type" and "pentagonal basket-type". It remains unclear why you think this is a useful classification. All clathrin coated vesicles are expected to have both hexagons and pentagons, and therefore, observing one or the other on the vesicle surface would not be informative. You need to indicate how the lattice is built in the text (always 12 pentagons with different numbers of hexagons). In keeping with this, your data indicate that the "pentagonal pits" are smaller in diameter, likely because they have fewer hexagons. This needs to be clarified also.

We believe that the classification of clathrin coated structures (CCSs) is useful as it provides an insight into the type of arrangements of clathrin used in plant cells, something which has never been examined before. We constantly found that these distinct basket formations exist. However, the exact physiological significance of this remains unclear and highlights the need for further work to define the precise structure of plant CCVs. We have added a few sentences to make these points clearer, Results paragraph three and Discussion subsection “Key characteristics of plant CME”.

In detail, the main reason we focused on the overall surface view shape of the basket (illustrated in Figure 1B), and not the arrangements of individual polygons, is because with current plant EM protocols we are unable to produce EM images where we can clearly visualize all the clathrin polygons within single fully formed CCVs. Further to this, the replica approach allows the classification of the basket arrangements/formations in situ which has the benefit of examining CCVs in a somewhat physiological environment. This is an advantage as it is well documented that CCVs in cells or tissues are substantially more heterogeneous in size and shape than the coats assembled for cryoEM studies (Cheng et al., 2007; Heymann et al., 2013; Kirchhausen et al., 2014). This is an important consideration, as the vesicle size and shape are affected by both membrane tension and specific adaptor proteins (which is especially interesting as we are currently unaware of what the plant adaptors are at single CME events are and further to this, TPLATE (a plant adaptor absent in mammalians and yeasts (Zhang et al., 2015)) and AP2 only have a 50% rate of co-localization (Gadeyne et al., 2014)) (Saleem et al., 2015). During the examination of the plant cell replicas, we consistently found that the basket of the CCVs were actually quite similar, however it was apparent that they exist in different populations definable by their basket arrangement and size. Therefore, the basket classification is useful to show that different populations of clearly identifiable assembles exist in the plant cells and it provides a way to examine the membrane bending using the only plant ultrastructural method which produced large numbers of CCVs.

We have also added a few lines to include information about the lattice is made from 12 pentagons and how numbers of hexagons effect the size of the CCV in Results paragraph four.

The data showing that CLC2 labels 60% of CHC1 coated pits is a good addition, but now, acknowledging that there are two types of CHC's further complicates interpretation of the data. It would be more accurate if you can clearly state that the live cell imaging data can technically be applicable to only one type of clathrin. Or maybe CHC1 represents most of the clathrin in these cells? Further clarification of whether CLC2 can associate with both CHCs or just one should be explicit in the text so it is apparent how much of the clathrin in plant cells is represented in the live cell imaging.The issue of different CHC's doesn't change the conclusions about the actin association with pits, but it does change the interpretation of the uncoating results. Looking at CLC2 vs adaptors means that you could be looking at more than one type of clathrin (CHC1 or CHC2 with CLC2 bound to either CHC). Maybe CHC1 clathrin dissociates faster than CHC2 clathrin, so the residual CLC2 represents a different clathrin uncoating by a different mechanism. Again, the fact that the uncoating dynamics in plants appear different to yeast and mammals is still valid, but the possibilities of more than one type of clathrin with potentially different behaviors needs to be acknowledged and discussed in full to ensure that the reader understands the implications.

We agree that the rate of co-localization of CLC2 and the CHCs is indeed interesting. And as such, it is an area of ongoing research within our lab. Currently, there has been no detailed assessment of the rate of co-localization between all the different clathrin isoforms in plant cells and it has been not an ambition of this report. However, it has been shown that the two CHC isoforms are functionally redundant for each other (Kitakura et al., 2011) and mass spectroscopy data show that the CHCs can interact with each other and all the CLCs (Adamowski et al., 2018; Gadeyne et al., 2014), so from a genetical and biochemical point of view, there is not a reason to assume that different isoforms can have distinct mechanistic roles but this needs to be examined in detail. We have added a few sentences to address this issue.